# A regeneration-triggered metabolic adaptation is necessary for cell identity transitions and cell cycle re-entry to support blastema formation and bone regeneration

Ana S Brandão[1]*, Jorge Borbinha[1], Telmo Pereira[1], Patrícia H Brito[2], Raquel Lourenço[1], Anabela Bensimon-Brito[3], Antonio Jacinto[1]*

[1]CEDOC, NOVA Medical School, Universidade Nova de Lisboa, Lisbon, Portugal; [2]UCIBIO, Dept. Ciências da Vida, Faculdade de Ciências e Tecnologia, Universidade NOVA de Lisboa, Lisbon, Portugal; [3]INSERM, ATIP-Avenir, Aix Marseille Univ, Marseille Medical Genetics, Marseille, France

**Abstract** Regeneration depends on the ability of mature cells at the injury site to respond to injury, generating tissue-specific progenitors that incorporate the blastema and proliferate to reconstitute the original organ architecture. The metabolic microenvironment has been tightly connected to cell function and identity during development and tumorigenesis. Yet, the link between metabolism and cell identity at the mechanistic level in a regenerative context remains unclear. The adult zebrafish caudal fin, and bone cells specifically, have been crucial for the understanding of mature cell contribution to tissue regeneration. Here, we use this model to explore the relevance of glucose metabolism for the cell fate transitions preceding new osteoblast formation and blastema assembly. We show that injury triggers a modulation in the metabolic profile at early stages of regeneration to enhance glycolysis at the expense of mitochondrial oxidation. This metabolic adaptation mediates transcriptional changes that make mature osteoblast amenable to be reprogramed into pre-osteoblasts and induces cell cycle re-entry and progression. Manipulation of the metabolic profile led to severe reduction of the pre-osteoblast pool, diminishing their capacity to generate new osteoblasts, and to a complete abrogation of blastema formation. Overall, our data indicate that metabolic alterations have a powerful instructive role in regulating genetic programs that dictate fate decisions and stimulate proliferation, thereby providing a deeper understanding on the mechanisms regulating blastema formation and bone regeneration.

*For correspondence:
ana.brandao@nms.unl.pt (ASB);
antonio.jacinto@nms.unl.pt (AJ)

Competing interest: The authors declare that no competing interests exist.

## Editor's evaluation

The authors provide convincing evidence to show that injury induces activation of glycolysis during zebrafish adult tail fin regeneration. This early activation is crucial for osteoblast dedifferentiation and proliferation, which are required for blastema formation and tail fin regeneration. This important study will be of interest to a broad audience in the fields of regeneration and metabolic regulation of developmental processes.

## Introduction

The zebrafish (*Danio rerio*) is a well-established model to study vertebrate regeneration due to its capacity to efficiently regenerate multiple organs and complex tissues, such as the caudal fin (*Sehring and Weidinger, 2020*; *Antos et al., 2016*; *Marques et al., 2019*). Caudal fin regeneration is an epimorphic process dependent on the formation of a blastema, a transient structure composed of proliferative lineage-restricted progenitor cells that originate from mature cells of the uninjured tissue (*Kawakami, 2010*; *Poss et al., 2003*). One of the main components of the caudal fin are the segmented bone elements, or bony-rays (*Pfefferli and Jaźwińska, 2015*; *Marí-Beffa and Murciano, 2010*), which have been shown to regenerate from new osteoblasts (OBs) arising from dedifferentiation of mature OBs close to the amputation site. Mature OBs acquire the cellular properties of less differentiated cells or pre-osteoblasts (pre-OBs) (*Knopf et al., 2011*; *Sousa et al., 2011*; *Stewart and Stankunas, 2012*; *Tu and Johnson, 2011*), which proliferate and redifferentiate into mature bone-producing OBs, thereby reconstituting the fin skeletal tissue (*Stewart et al., 2014*; *Brandão et al., 2019*; *Brown et al., 2009*; *Wehner et al., 2014*). Mammals, in contrast to zebrafish, have a poor ability to regenerate (*Xia et al., 2018*; *Poss, 2010*; *Tanaka and Reddien, 2011*). Mammalian bone, in particular, has an intrinsic capacity to be remodeled throughout life and, to a certain extent, repair after fracture (*Salhotra et al., 2020*; *Raggatt and Partridge, 2010*), but it fails to fully regenerate. This is mainly achieved through activation of signalling cascades that culminate in the recruitment and differentiation of osteoprogenitors into OBs, the bone-forming cells. In this scenario, understanding how OBs and OB sources are activated and recruited in zebrafish can greatly contribute to new therapeutic solutions to improve bone formation and restoration in mammals (*Yao et al., 2013*). Activation of reprogramming events that allow cells to change fate in response to injury appear to be conserved biological processes, which are used to prompt tissue regeneration in several organisms (*Tanaka and Reddien, 2011*; *Tanaka, 2016*; *Jopling et al., 2011*), including in mammals (*Seifert and Muneoka, 2018*; *Johnson et al., 2020*). Therefore, it is crucial to identify the early molecular events inducing cell dedifferentiation to promote tissue regeneration.

Cell identity and functional state often reflect a specific metabolic profile that depends, for instance, on nutrient and oxygen availability, and on bioenergetics and biomass requirements (*Folmes et al., 2012*; *Ito and Suda, 2014*). Metabolic routes serve not only the crucial purpose of converting or use energy to maintain cellular integrity and survival, but also have a pivotal role in restructuring gene expression to determine cell identity and function by influencing cell signalling and epigenetic modulators (*Tarazona and Pourquié, 2020*; *Ly et al., 2020*). Glucose metabolism is currently regarded as a powerful instructor of cell fate decisions (*Wei et al., 2018*; *Tatapudy et al., 2017*; *Ghosh-Choudhary et al., 2020*) during development. It is well documented that stem cells and their differentiated progeny have distinct metabolic preferences: stem cells often favour non-oxidative glycolysis, while differentiated non-dividing somatic cells rely on mitochondrial oxidative phosphorylation (OXPHOS) (*Shyh-Chang and Daley, 2015*; *Panopoulos and Izpisua Belmonte, 2011*; *Tsogtbaatar et al., 2020*; *Mathieu and Ruohola-Baker, 2017*). Importantly, the metabolic signature is not static and can rapidly switch according to the cellular demands, a phenomenon commonly designated as metabolic reprogramming (*Folmes et al., 2012*; *Ghosh-Choudhary et al., 2020*). This is particularly relevant under certain environmental conditions when tissue homeostasis is breached, such as during inflammation, and during disease. Both activated T cells and pro-inflammatory macrophages switch metabolic profiles compatible with a prevalence of aerobic glycolysis (*Shyer et al., 2020*; *Viola et al., 2019*). The shift to aerobic glycolysis is also observed in cancer cells, where it is referred as the Warburg effect (*DeBerardinis et al., 2008*; *Intlekofer and Finley, 2019*; *Warburg et al., 1927*; *Liberti and Locasale, 2016*; *Vander Heiden et al., 2009*).

Despite the considerable amount of information on the role of metabolism during development and disease (*Vander Heiden et al., 2009*; *Lee et al., 2017*; *DeBerardinis and Thompson, 2012*; *Tzika et al., 2018*; *Bettencourt and Powell, 2017*), the link between glucose metabolism and regeneration is still far from understood. Shifts in metabolism leading to a prevalence of glycolysis have already been observed in planarians and amphibians during regeneration (*Osuma et al., 2018*; *Love et al., 2014*; *Alibardi, 2014*; *Varela-Rodríguez et al., 2020*). More recently, glycolysis was proposed to modulate specific aspects of regeneration in the zebrafish, specifically in the larval tail (*Sinclair et al., 2021*) and in the heart (*Honkoop et al., 2019*; *Fukuda et al., 2020*). Nevertheless, much

remains to be elucidated on how metabolism influences changes in cell identity necessary to promote tissue regeneration.

Here, we use the adult zebrafish caudal fin to determine the role of energy metabolism in mature OB dedifferentiation and pre-OB recruitment. By performing a series of gene expression and metabolomic studies, we observed that changes in the metabolic signature triggered upon amputation are at the core of the initial set-up of the cellular programmes that control caudal fin regeneration. These data indicate that mature OBs, and possibly other cell lineages in the caudal fin, undergo a metabolic adaptation that increases glycolysis. We further demonstrate that these changes in metabolism are necessary for mature OBs to commit to a specialized genetic program that enables them to dedifferentiate, proliferate and act as progenitor cells, ensuring proper bone regeneration. Taken together, our results demonstrate that metabolic reprogramming is one of the earliest described cellular events that dictate adult caudal fin and bone regeneration.

## Results

### Osteoblast dedifferentiation occurs before blastema formation and during the wound healing phase

Caudal fin bone regeneration is achieved through activation of cell sources, such as mature OBs, that change their identity to generate new OBs. Under homeostatic conditions, mature OBs reside as quiescent bone-lining cells, but lose their differentiated character when undergoing dedifferentiation (*Knopf et al., 2011*; *Sousa et al., 2011*). This process is characterized by the downregulation of mature markers, such as *bone gamma-carboxyglutamic acid-containing protein* (*bglap*) at 12 hpa (*Knopf et al., 2011*; *Sousa et al., 2011*; *Long, 2011*), and upregulation of the pre-OB marker, *runx2*, at 24 hpa (*Knopf et al., 2011*; *Sousa et al., 2011*; *Salhotra et al., 2020*). OBs undergoing dedifferentiation detach from the bony-ray surface via an EMT-like event, migrate toward the stump region, re-enter the cell cycle and form a pool of pre-OB cells that incorporates the blastema (*Figure 1A*; *Knopf et al., 2011*; *Sousa et al., 2011*; *Stewart et al., 2014*). To determine if OBs show signs of dedifferentiation in early stages of regeneration, we increase the time resolution of the initial OB response to amputation. First, we analyze the relative expression of mature (*bglap*) and pre-OB (*runx2a, runx2b*) markers 6 hr post-amputation (hpa) in regenerating caudal fins in relation to uninjured caudal fins (0 hpa). We observed a downregulation of *bglap* and upregulation of *runx2a* (*Figure 1B*) suggesting that, in contrast to previous studies (*Knopf et al., 2011*; *Sousa et al., 2011*), mature OBs are already undergoing transcriptional changes consistent with dedifferentiation as early as 6 hpa. We then analyzed the *bglap*:EGFP transgenic line which, due to the stable GFP signal, allows to follow mature OB for long periods even upon *bglap* downregulation (*Sousa et al., 2011*). Using live-imaging approaches, we monitored the two segments below the amputation plane of each bony-rays and performed a time-course analysis of OB migration every 5 hr during the first 25 hpa (*Figure 1C and D*). Tracking of *bglap*-positive (Bglap+) OBs revealed that OBs from the segment below amputation (segment 0) become motile at the 5–10 hpa time-interval and in average reach the amputation plane around 20–25 hpa (*Figure 1C and D*), while Bglap +OB from the second segment (segment-1) remain predominantly immotile (*Figure 1C*). Subsequently, we assessed when OBs acquire proliferative capacity. Immunofluorescence for PCNA (marker of G1 phase) indicated that the proportion of Bglap +OBs entering the cell cycle progressively increases, specifically from 12 to 24 hpa (*Figure 1E*). At 24 hpa, almost 80% of Bglap +OBs in the first segment have entered the G1 phase (*Figure 1E*). To examine when OBs begin to show signs of dedifferentiation toward a pre-OB state, we performed immunofluorescence for Runx2 in *bglap*:EGFP transgenic fins. In homeostasis conditions (0 hpa), Runx2 is observed in the nucleus of bone-lining mature OB (Runx2 +Bglap + ; *Figure 1F, F' and H*), which is in accordance with Runx2 being expressed at basal levels in fully differentiated OB (*Conaway et al., 2013*). At 12 and 24 hpa, we observed a slight decrease in the total number of Runx2 +Bglap + OB population within the first segment below amputation (*Figure 1G, G' and H*). OBs are not described to undergo apoptosis at this stage (*Knopf et al., 2011*), suggesting that decrease in the Runx2 +Bglap + OBs could be due to actual loss of the *bglap* differentiation marker in a portion of dedifferentiated OB. Nevertheless, remaining Runx2 +Bglap + possess higher levels of Runx2 when compared to Runx2 +Bglap + OBs at 0 hpa (*Figure 1F–G'* arrows), further indicating that they converted into a pre-OB phenotype, in accordance with published data (*Mishra et al., 2020*). In addition, at 0 hpa a small number of

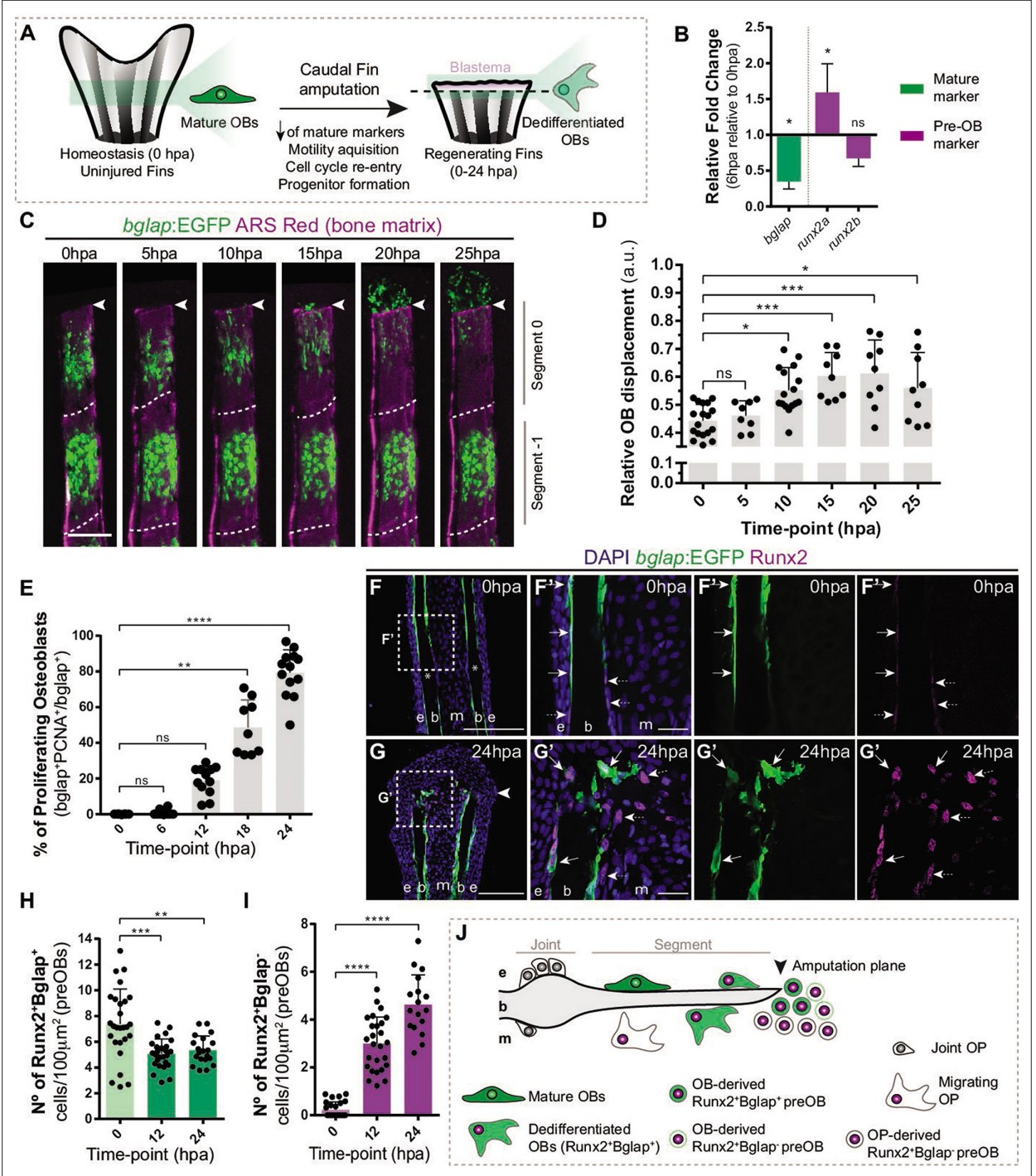

**Figure 1.** Osteoblast dedifferentiation time-window during caudal fin regeneration. (**A**) Biological traits of OB dedifferentiation process. (**B**) Relative gene expression of mature (green) and pre-OB (magenta) markers, at 6 hpa relative to 0 hpa. Statistical analysis on graph corresponds to paired t-test with Welch's correction. Mean ± SD are displayed (n=4 biological replicates). (**C**) Live imaging analysis of OB motility in bglap:EGFP fish (green) during the first 25 hpa, highlighted in the segment bellow amputation (segment 0) and segment –1. Bony-rays are labeled with Alizarin red (magenta). White dashed lines delineate the intersegment region. (**D**) Quantification of the relative OB displacement in segment 0. Statistical analysis displayed on graph corresponds to Kruskal-Wallis test with Mean ± SD (n=9–18 bony-rays). (**E**) Percentage of proliferating OBs through immunofluorescence against PCNA in bglap:EGFP fish. Statistical analysis displayed on graph corresponds to Kruskal-Wallis test with Mean ± SD (n=9–13 cryosections). (**F-G'**) Representative cryosection images of bglap:EGFP (green) fins immunostained for Runx2 (magenta) and counterstained with DAPI (blue), in (**F, F'**) uninjured fish and (**G, G'**) at 24 hpa; arrows indicate Runx2 +Bglap + cells and dashed arrows indicate Runx2 +Bglap cells. (**H, I**) Quantification of (**H**) Runx2 +Bglap + and (**I**) Runx2 +Bglap cells during the first 24hpa. Statistical analysis displayed on graph corresponds to Mann-Whitney test

*Figure 1 continued on next page*

*Figure 1 continued*

with Mean ± SD (n=18–27 cryosections). (**J**) Cellular sources that contribute for new pre-OBs formation after injury include mature osteoblasts and potentially joint OP. White arrowhead indicates amputation plane and dashed squares represent magnified panels in F' and G'. E: epidermis; b: bone; m: mesenchyme; ns: not significative; *p<0,05; **p<0,01; ***p<0,001; ****p<0,0001. Scale bars represent 100 μm and 30 μm in magnified panels. See *Figure 1—source data 1*.

The online version of this article includes the following source data for figure 1:

**Source code 1.** MatLab scripts quantify the relative osteoblast displacement after caudal fin amputation.

**Source data 1.** Spreadsheets detailing the results regarding the characterization of osteoblast dedifferentiation through caudal fin regeneration, specifically (**B**) the relative gene expression analysis, (**E**) the percentage of proliferating osteoblast, and (**H,I**) the quantification of Runx2+Bglap + and Runx2+Bglapcells during the first 24hpa.

pre-OB (Runx2 +Bglap-) was detected in the bony-rays intersegment/joint (*Figure 1F and F'* asterisk,I), which may correspond to a population of OB progenitors (OPs) recently identified in this region, refered as joint OP (*Ando et al., 2017*). At 12 and 24 hpa, we observed an increase in the number of Runx2 +Bglap cells, suggesting that pre-OB arise in the first 12 hpa and their numbers increase until the beginning of blastema formation at 24 hpa (*Figure 1F-G' and I*). Altogether, these data indicate that mature OBs lose their differentiated character and contribute to the pre-OB pool in a time-window between 12 and 24 hpa. Nevertheless, we cannot exclude that pre-OB can also arise from the joint-associated OB progenitor niche. Thus, we hypothesize that at 24 hpa, multiple OB sources are recruited to contribute to the blastema (*Figure 1J*).

Here, we show that mature OB dedifferentiation is an early response to injury and entails important transcriptional and phenotypic alterations, occurring in a narrow time-window between 6–12 hpa, before blastema induction and during the wound healing phase.

## Metabolic adaptation toward glycolysis occurs at early stages of fin regeneration prior to blastema formation

Having established that the first hours after caudal fin amputation are crucial for the activation of OBs, we proceeded with the identification of the initial regulators of OB dedifferentiation. We isolated OBs from the first bony-ray segment below amputation at 0 hpa, our control population and closest to a homeostatic state, and at 6 hpa, when most dedifferentiation features are not detected yet, and performed a genome-wide transcriptomic analysis (zebrafish 8x60 K ArrayXS technology) (*Figure 2—figure supplement 1A*). OBs were isolated by FACS using the *bglap*:EGFP transgenic line (*Figure 2—figure supplement 1B*). By comparing the expression profiles of mature OBs (0 hpa) and dedifferentiating OBs (6 hpa), we evaluated whether our set of differentially expressed (DE) genes was associated with a specific biological process or signalling pathway, particularly relevant for OB dedifferentiation. We observed that OBs show a dynamic transcriptional response at 6 hpa in comparison to OBs from uninjured conditions with almost 2200 differentially expressed genes, from which 1040 were downregulated and 1130 were upregulated (*Figure 2—figure supplement 1C*,D). These data further demonstrates that the dedifferentiation machinery is triggered very early during regeneration (for more details about the set of DE genes see *Supplementary file 1*). Importantly, a large set of genes related to energy metabolism was also dramatically altered (*Supplementary file 1*).

Depending on the energy and biomass demands, cells can uptake glucose and, through glycolysis, use it to produce pyruvate, which can serve as a substrate to: generate acetyl-CoA and fuel mitochondrial OXPHOS, obtaining a high energy yield; or to produce lactate, allowing diversion of metabolic intermediates from glycolysis toward various biosynthetic pathways (*Lunt and Vander Heiden, 2011*; *Rabinowitz and Enerbäck, 2020*; *Figure 2A*). From our transcriptome data set of DE, we observed that at 6 hpa OBs upregulate the expression of major glycolytic enzymes, such as *hexokinase 1* (*hk1*) and *phosphofructokinase* (*pfkp*), whereas OXPHOS components remain mostly unchanged (*Figure 2B*, *Supplementary file 1*). Most importantly, we detected a significant increase in *lactate dehydrogenase* (*ldha*) expression, suggesting an increase in lactate production, possibly by diverting pyruvate from mitochondrial oxidation. (*Figure 2B*, *Supplementary file 1*). These data suggest that dedifferentiating OBs change their metabolic signature, adopting a metabolic program that increases glycolysis. Since OBs only account for around 1–3% of the total cell number in a bony-ray segment (*Figure 2—figure supplement 1B*), we evaluated whether these changes in gene expression were

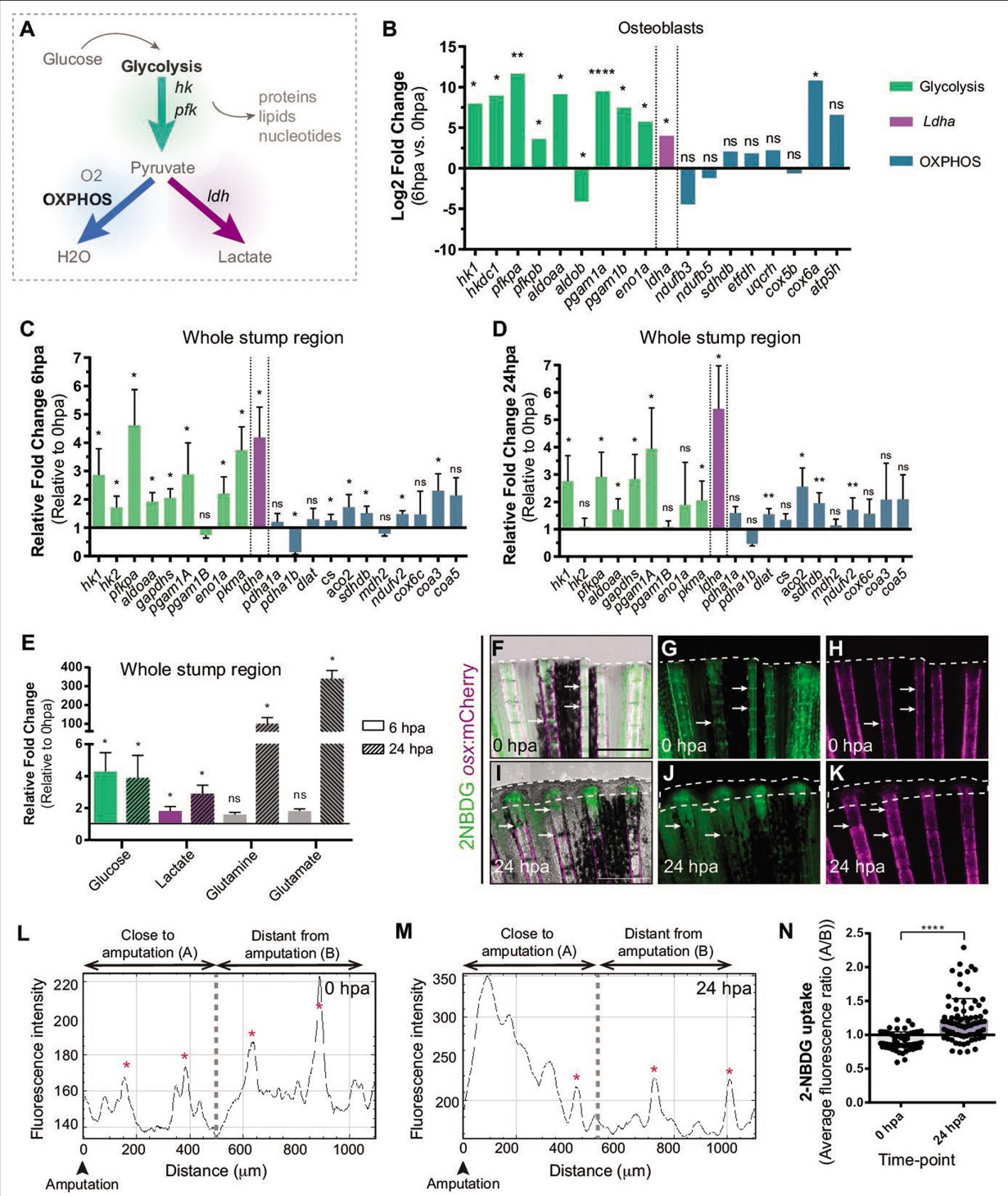

**Figure 2.** Metabolic adaptation is triggered during zebrafish caudal fin regeneration. (**A**) Schematic representation of glucose metabolism. (**B**) OB gene expression profile of glycolytic enzymes (green), ldha (magenta) and OXPHOS components (blue) at 6 hpa relative to uninjured conditions (0 hpa) obtained from the OB ArrayXS. The horizontal axis represents the log2 fold-change and p-values on a negative log10 scale. Statistical analysis with t test and Welch's correction (n=3 biological replicates), Mean ± SD are displayed. (**C, D**) Relative gene expression of glycolytic enzymes (green), ldha (magenta) and OxPhos components (blue), in the whole fin stump, at (**C**) 6 hpa and at (**D**) 24 hpa in comparison to uninjured conditions (0 hpa). Statistical analysis with paired t test (n=5 (**C**) and 4 (**D**) biological replicates). (**E**) Metabolite measurements at 6 hpa (clean columns) and 24 hpa (stroked columns) in relation to uninjured conditions (0 hpa), in the whole fin stump. Statistical analysis with Mann-Whitney test (n=4 biological replicates). (**F–K**) Live imaging of 2NDBG uptake (green) in osx:mCherry fish (magenta) at (**F–H**) 0 hpa and (**I–K**) 24 hpa. Arrows indicate uptake of 2NBDG in the

*Figure 2 continued on next page*

*Figure 2 continued*

intersegment regions. White dashed line delineates the regenerated tissue. Scale bar represents 500 µm. (**L–M**) Intensity of 2NBDG uptake in regions close and distant to the amputation site, at (**L**) 0 hpa and (**M**) 24 hpa. Red * indicate peaks of 2NDBG uptake in the intersegments. (**N**) Ratio of 2NBDG uptake at 0 hpa and 24 hpa. Statistical analysis on graph corresponds to Mann-Whitney test. Mean ± SD are displayed (n=54 and 83 bony-rays). ns: not significative; *p<0,0001. See *Figure 2—source data 1*.

The online version of this article includes the following source data and figure supplement(s) for figure 2:

**Source data 1.** Spreadsheets detailing the results of the metabolic adaptation, specifically the quantification of (**C,D**) the relative gene expression analysis, (**E**) the relative metabolite levels, and (**N**) the average fluorescent levels of 2-NBDG.

**Figure supplement 1.** Isolation and gene expression analysis osteoblasts undergoing dedifferentiation.

**Figure supplement 2.** Changes in metabolism are accompanied by alterations in mitochondria dynamics.

**Figure supplement 2—source data 1.** Spreadsheets detailing the results of the mitochondrial dynamics, specifically (**A,B**) the relative gene expression analysis, and the quantification of (**I**) the number of mitochondria per cell and (**J**) the percentage of each mitochondria volume.

specific to OB undergoing dedifferentiation or are also observed as a general behavior induced upon injury. For that, we collected the first segment below the amputation and analyzed by qPCR the expression profile of key glycolytic and OXPHOS genes in regenerating caudal fins from 6 hpa and 24 hpa and compared to control uninjured caudal fins (0 hpa). At 6 hpa, when cells start to dedifferentiate, most glycolytic enzymes were highly upregulated, including *hk1*, *hk2*, *pfkpa*, and *pyruvate kinase* (*pkma*) (*Figure 2C*). Importantly, as observed during OB dedifferentiation, *ldha* expression was increased and *pyruvate dehydrogenase a 1b* (*pdha1b*) was downregulated. *pdha1b* is part of the Pdh1 complex, which catalyzes irreversibly the conversion of pyruvate to acetyl-CoA and progression to OXPHOS, further corroborating that at this stage pyruvate is being shunt from the mitochondria and stimulating lactate-producing glycolysis (*Figure 2C*). As for the OXPHOS components analyzed, most remained unchanged or slightly upregulated. However, this increase in expression was not as accentuated as the one observed for the glycolytic pathway (*Figure 2C*). Later, at 24 hpa, when most cells have dedifferentiated and the blastema starts to be assembled, we observed that most glycolytic enzymes were still significantly upregulated, although not as striking as at 6 hpa time-point, and *ldha* expression continued to be upregulated (*Figure 2D*). At this time-point, we also observed an increase in expression of some OXPHOS components (*Figure 2D*), suggesting that blastema formation may also require mitochondrial glucose oxidation. Consistent upregulation of the genes associated with glycolysis and lactate production suggests that the OB dedifferentiation process and possibly the early response to amputation are characterized by a metabolic adaptation toward glycolysis.

Nevertheless, gene expression data may not reflect an actual activation of a specific metabolic pathway. Therefore, to corroborate these observations, we used mass spectrometry (MS) to quantify specific metabolites and determine the prevalent energy metabolism route. For that, we dissected the whole fin stump and analyzed the amount of glucose, lactate, glutamine and glutamate at 0, 6, and 24 hpa. According with our transcriptome data, we observed an increase in glucose and lactate at 6 and 24 hpa in relation to control fins (0 hpa), which is consistent with an increment in the glycolytic influx (*Figure 2E*). We also observed a strong increase of glutamine and glutamate at 24 hpa but not at 6 hpa (*Figure 2E*). Glutamine and glutamate act as important substrates for protein and nucleotide synthesis necessary to support cellular integrity and growth suggesting an increase in biosynthesis (*Newsholme et al., 2003*; *Yoo et al., 2020*).

We also monitored glucose uptake during regeneration using a fluorescent glucose analogue (2-NBDG). In control fins (0 hpa), we detected glucose uptake in the intersegment region (*Figure 2F–H* arrows,L,N), where a population of OP have been identified (*Ando et al., 2017*), which may imply that under homeostatic conditions, these cells require higher levels of glucose to maintain the progenitor niche. At 24 hpa, we observed a significant increase of glucose uptake in the blastema primordium and in the first segment below the amputation (*Figure 2I–K, M and N*).

Finally, metabolic changes are usually associated to mitochondria dynamics events, fusion and fission, which control mitochondria shape, distribution and size (*van der Bliek et al., 2013*). A balance between fission and fusion allows mitochondria to acquire the morphological structure to conduct specific cellular requisites and functions. Fusion maximizes OXPHOS for energy production in differentiated cells by stabilisation of the mitochondrial respiratory network, characterized by long mitochondria chains. In contrast, fission generates smaller and spherical mitochondria and is usually associated

with stemness and with cells that rely on a glycolytic metabolism (*Wanet et al., 2015*; *Lisowski et al., 2018*; *Prigione et al., 2015*). Considering the increase in the glycolytic influx previously observed, we decided to address how mitochondria were affected during regeneration. By performing a qPCR analysis at 0, 6, and 24 hpa, we observed an increase in expression of genes associated with mitochondrial fission, namely *drp1* and *fis1*, at 6 hpa but not at 24 hpa, when compared to uninjured condition (0 hpa) (*Figure 2—figure supplement 2*). By quantifying the number of mitochondria using the reporter line MLS-GFP, which labels the mitochondria membrane (*Kim et al., 2008*), we noticed an increase in the number of mitochondria per cell at 6 and 24 hpa, in relation to uninjured control (*Figure 2—figure supplement 2C-H',I*). To address if there were also changes in mitochondria size that could correspond to an actual increase in fission, we measured mitochondria volumes and grouped them into different size intervals. Mitochondria usually measure between 0.5 and 3 $\mu m^3$, but their size and shape can vary considerably (*Shami et al., 2021*; *Wiemerslage and Lee, 2016*). In accordance with an increase in mitochondria fission, we observed a significant increase in percentage of mitochondria smaller than 0,1 $\mu m^3$ at 6 hpa, while at 24 hpa the mitochondria size profile is similar to the 0 hpa condition (*Figure 2—figure supplement 2C-H',J*). Taken together, these results are in consistent with an increase in mitochondrial fission triggered by amputation and further supports the hypothesis that the early stages of regeneration involve an upregulation of glycolytic activity, which is accompanied by a temporally restricted program of mitochondria fission.

Thus, metabolic adaptation enhancing glycolysis seems to constitute an integral response to amputation, tightly regulated in terms of time and space. Our transcriptome and metabolome studies indicate a specific metabolic signature as part of an early response to amputation by OBs and by the remaining caudal fin tissue. Overall, cells respond to injury by increasing the glycolytic influx, lactate production and mitochondrial fission just preceding dedifferentiation and blastema formation. These results corroborate the hypothesis that amputation induces a reprogramming of the metabolic profile, which may be part of a general program that coordinates the regenerative response.

## Glycolytic influx and lactate generation supports blastema formation during caudal fin regeneration

To investigate the functional relevance of the energy metabolism, we chose to start with a broader analysis in which we manipulated specific branches of energy metabolism with pharmacological compounds and address its requirement for blastema formation (*Figure 3A*). The blastema is a hallmark and prerequisite of epimorphic regeneration and is fully assembled within 48 hpa (*Kawakami, 2010*; *Poss et al., 2003*; *Pfefferli and Jaźwińska, 2015*). Firstly, we inhibited glycolysis using an established glucose analogue, 2-Deoxy-D-glucose (2DG) (*Sinclair et al., 2021*; *Honkoop et al., 2019*), that competes with glucose for Hk catalytic domain, the first step of glycolysis, inhibiting the downstream products derived from glucose. 2DG was administered in different time points right after amputation to affect different stages of blastema formation: for a short period before blastema formation (0 hpa and 0-12hpa), or for a prolonged period, which includes the blastema assembly phase (0-24hpa and 0-36hpa). Fins were imaged at 48 hpa to determine the effect of 2DG on blastema growth by measuring the total fin regenerated area and quantifying the percentage of regenerated area, in relation to control caudal fin regenerated area (*Figure 3B*). We observed that the higher the duration of the treatment the stronger the effect on blastema growth in comparison to control fins (*Figure 3C and D–K*). Interestingly, a single dose of 2DG administered at 0 hpa already showed significant smaller regenerates when compared to control fins (*Figure 3C–E*). Administration of 2DG between 0–12, 0–24, and 0–36 hpa had a dose-dependent inhibitory effect on the overall regenerated area (*Figure 3C and F–K*). Accordingly, blocking of the glycolytic influx for the first 36 hpa led to a complete abrogation of blastema assembly (*Figure 3C and J–K*). Similarly, the glycolysis inhibitor acting downstream of 2DG, 3-(3-pyridinyl)–1-(4-pyridinyl)–2-propen-1-one (3PO), which partially inhibits the glycolytic activator of Pfk (*Schoors et al., 2014 Figure 3A*), also exhibited a general impairment of caudal fin regeneration, as observed at 48 hpa in relation to control fins (*Figure 3—figure supplement 1A-D*).

Next, we blocked the conversion of pyruvate to lactate using the inhibitor of Ldh, Sodium Oxamate (S.O.) (*Fiume et al., 2010*; *Figure 3A*). S.O. administration during the first 36 hpa resulted in a decrease of the caudal fin regenerated area in respect to control caudal fins (*Figure 3L–O*), although exhibiting a milder effect when compared to glycolysis inhibition (*Figure 3C–K* and *Figure 3—figure supplement 1A-D*). In contrast, inhibition of OXPHOS using a mitochondrial pyruvate carrier inhibitor

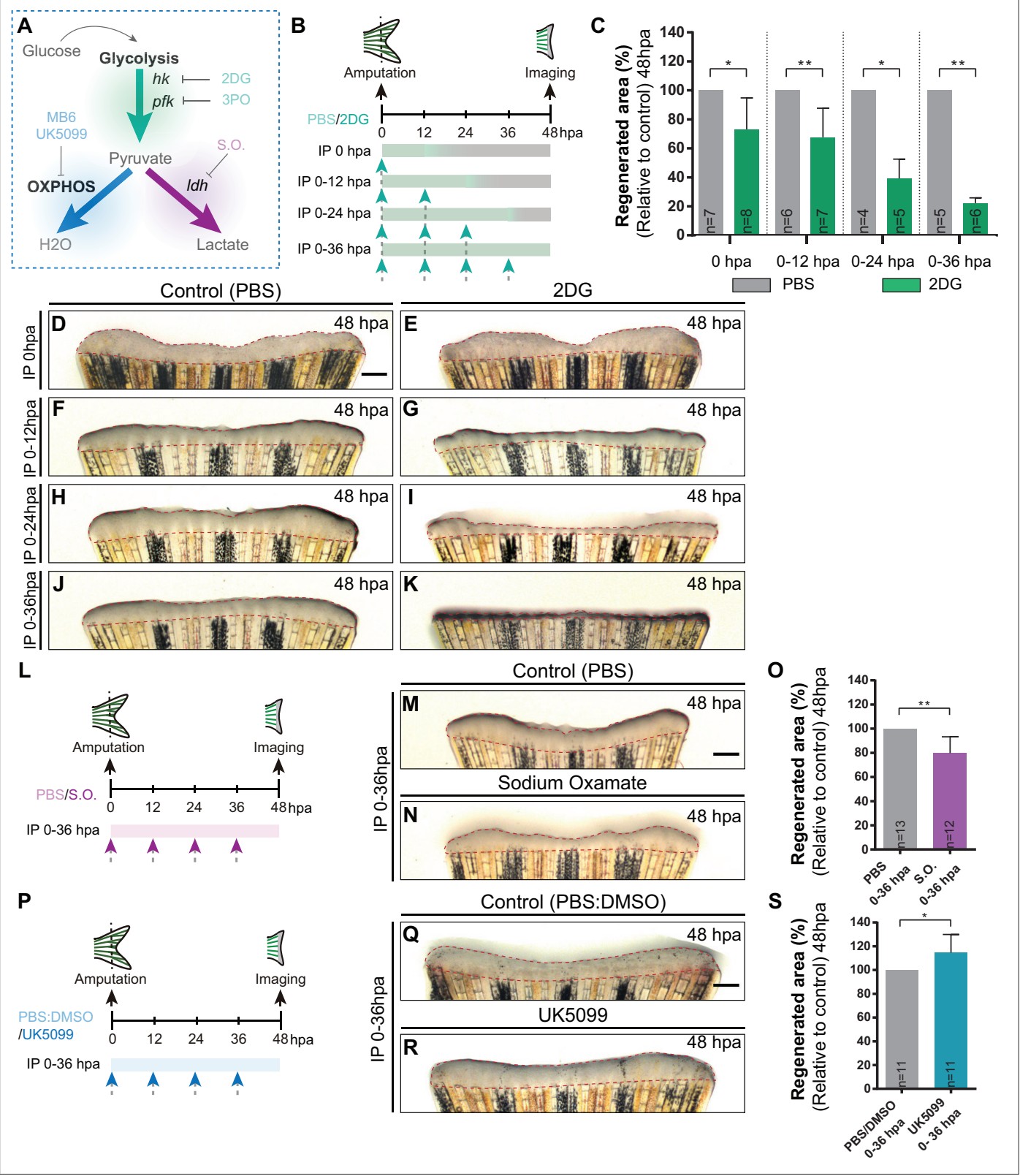

**Figure 3.** Inhibition of glycolysis, but not OXPHOS, impairs blastema formation. (**A**) Schematic representation of the compounds used to manipulate glucose metabolism. (**B**) Experimental design used to inhibit the glycolytic influx during fin regeneration. Control and treated fish are administered, via IP injection, with vehicle (PBS) or glycolytic inhibitor, 2DG, respectively, every 12 hr, from fin amputation (0 hpa) until 48 hpa. Different time-intervals were used for injections: (0 hpa) IP injection at (0 hpa; 0–12 hpa) IP injection at 0 and 12 hpa; (0–24 hpa) IP injection at 0, 12, and 24 hpa; (0–36 hpa) IP

*Figure 3 continued on next page*

*Figure 3 continued*

injection at 0, 12, 24, and 36 hpa. (**C**) Quantification of the total fin regenerated area at 48 hpa, after vehicle (PBS) or 2DG injection, at specific time-intervals during regeneration. (**D–K**) Representative images of 48 hpa fins treated with (**D,F,H,J**) vehicle (PBS) or (**E,G,I,K**) 2DG during different time-intervals. (**L**) Experimental design used to inhibit the lactate formation during fin regeneration. Fish are administered, via IP injection, with vehicle (PBS) or S.O. every 12 hr, from fin amputation (0 hpa) until 48 hpa. (**M, N**) Representative images of 48 hpa caudal fin treated with (**M**) vehicle (PBS) or (**N**) S.O. (**O**) Quantification of the total fin regenerated area at 48 hpa, after vehicle (PBS) or with S.O. injection. (**P**) Experimental design used to inhibit pyruvate translocation to mitochondria during fin regeneration. Fish are administered, via IP injection, with vehicle (PBS) or MPC inhibitor, UK5099, every 12 hr from fin amputation (0 hpa) until 48 hpa. (**Q, R**) Representative images of 48 hpa fins treated with (**Q**) PBS:DMSO (control) or (**R**) UK5099. (**S**) Quantification of the total fin regenerated area at 48 hpa, after vehicle (PBS) or with UK5099 injection. For all graphs, statistical analysis corresponds to Mann-Whitney test with Mean ± SD, sample number is displayed on each column and corresponds to single fish. Scale bar represents 500 µm. Dashed lines define the regenerated tissue. * $p<0.05$, ** $p<0.01$. See *Figure 3—source data 1*.

The online version of this article includes the following source data and figure supplement(s) for figure 3:

**Source data 1.** Spreadsheets detailing the results of glycolysis and OXPHOS inhibitory assays, specifically the quantification of (**C**) the percentage of regenerated fin area after 2DG treatment, (**O**) the percentage of regenerated fin area after S.O. treatment, and (**S**) the percentage of regenerated fin area after UK5099 treatment.

**Figure supplement 1.** Inhibition of glycolysis impairs blastema formation.

**Figure supplement 1—source data 1.** Spreadsheets detailing the results of alternative glycolysis and OXPHOS inhibitory assays, specifically the quantification of the percentage of the regenerated fin area after (**D**) 3PO and (**H**) MB6 treatment.

(UK5099) (*Hildyard et al., 2005*; *Figure 3A*) led to a mild, although significant, increase in the regenerated area at 48 hpa when compared to control fins (*Figure 3P-S*), suggesting an improvement of the regenerative ability. The same effect was not observed using an inhibitor of the mitochondrial electron transport chain (MitoBlock-6 (MB6)) (*Dabir et al., 2013*; *Figure 3A*; *Figure 3—figure supplement 1E-H*). Together, these results indicate that glycolysis and lactate generation, but not OXPHOS, are essential for the initial stages of caudal fin regeneration. Our data strongly supports the hypothesis that the cells that respond to amputation reprogram their metabolic profile since very early thereby stimulating the glycolytic machinery/influx to support blastema formation.

## Inhibition of glycolysis interferes with osteoblast dedifferentiation and pre-osteoblast pool assembly and proliferation

Considering that glycolysis inhibition culminated in a severe impairment of blastema assembly, we investigated whether those defects could be due to a role of glycolysis in mediating cell dedifferentiation and/or re-acquisition of proliferative capacity (*Figure 4A*), processes that precede and are indispensable for blastema formation. Therefore, we blocked glycolysis with 2DG and performed a detailed characterization of OB dedifferentiation at 24 hpa (*Figure 4B*). We began by analyzing the expression profile of OB markers that serve as a read-out of their dedifferentiated status, namely *bglap* and *runx2*. We observed that inhibition of the glycolytic influx led to a significant upregulation of *bglap* and to a decrease in both *runx2* orthologues (*Figure 4C*), as opposed to what happens in OB undergoing dedifferentiation in normal regenerating conditions (*Knopf et al., 2011*; *Sousa et al., 2011*). Immunofluorescence analysis for Runx2 in *bglap*:EGFP reporter fish, showed that 2DG treatment had no effect in the number of dedifferentiated Runx2 +Bglap + pre-OBs (*Figure 4D–G*). However, the number of Runx2 +Bglap- pre-OBs was significantly reduced (*Figure 4E–G*) when compared to control fins (*Figure 4D, F and G*). Overall, these data indicate that blocking glycolytic influx hampers mature OB dedifferentiation, which become unable to operate as a source of pre-OB resulting in impaired pre-OB pool assembly within the blastema primordium.

To further validate our observations, we decided to evaluate whether the pathways proposed to mediate mature OB dedifferentiation where also altered upon glycolysis inhibition. During homeostasis NF-kB pathway maintains retinoic acid (RA) signalling in OBs, supporting differentiation. Upon amputation, NF-kB pathway becomes inactivated and RA is degraded through activity of the RA-degrading enzyme, Cyp26b1, thereby inducing OB dedifferentiation (*Mishra et al., 2020*; *Blum and Begemann, 2015*; *Figure 4A*). In addition, Bmp signaling is also considered to be a potent inducer of OB differentiation during regeneration (*Stewart et al., 2014*; *Brandão et al., 2019*; *Smith et al., 2006*; *Quint et al., 2002*). Thus, we performed qPCR analysis at 24 hpa of NF-kB target genes (*e.g. nfkbiaa* and *nfkbiab*), retinoic acid degrading enzyme (*e.g. cyp26b1*), and Bmp ligands (*e.g. bmp2a* and *bmp2b*) in control and 2DG-treated fins (*Figure 4C*). We observed an increase in NF-kB target

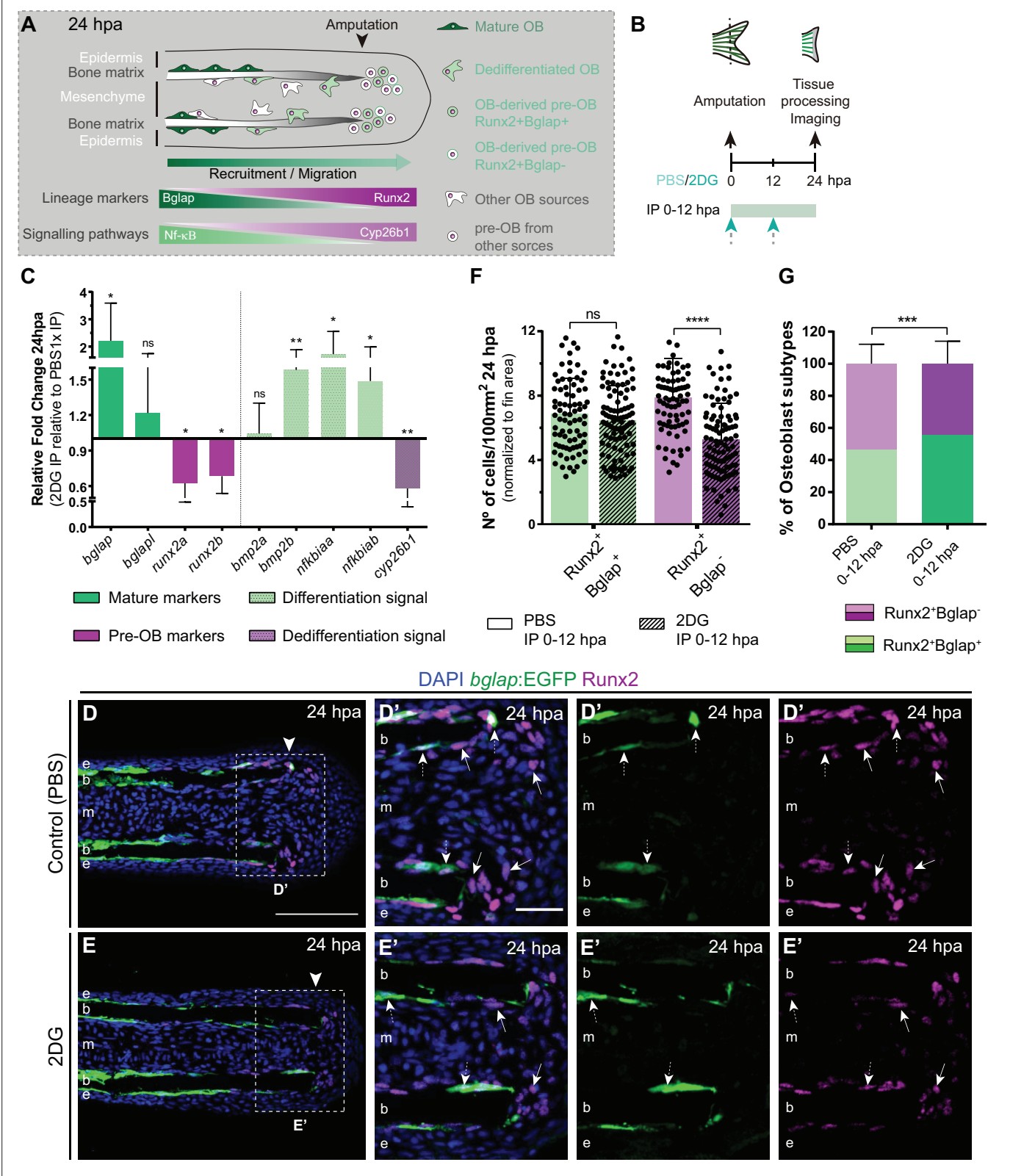

**Figure 4.** Inhibition of glycolysis impairs osteoblast dedifferentiation. (**A**) Schematic representation of pre-OBs formation during regeneration. Pre-OBs arise from OB dedifferentiation and potentially from the joint OP niche. OB dedifferentiation is correlated with inactivation of NF-ΚB and increase in Cyp26b1 activity. (**B**) Experimental design used to inhibit glycolysis. Fish are administered, via IP injection, with vehicle (PBS) or 2DG, from fin amputation (0 hpa) until 24 hpa. (**C**) Relative gene expression of mature and pre-OBs markers, and differentiation and dedifferentiation pathways,

*Figure 4 continued on next page*

*Figure 4 continued*

in the whole fin stump at 24 hpa, in 2DG treated fins compared to control condition (0 hpa). Statistical analysis with paired t-test (n=10 biological replicates). (**D-E'**) Representative cryosection images of 24 hpa bglap:EGFP (green) caudal fins immunostained for Runx2 (magenta) and counterstained with DAPI (blue), in fish treated with (**D,D'**) vehicle (PBS) or (**E,E'**) 2DG. White dashed boxes delineate magnified panels in D' and E'. Arrows indicate Runx2 +Bglap cells. Dashed arrows indicate Runx2 +Bglap + cells. Arrowhead indicates amputation plane. E: epidermis; b: bone; m: mesenchyme. Scale bar represents 100 µm and 30 µm in magnified panels. (**F**) Total number of Runx2 +Bglap + and Runx2 +Bglap cells per area at 24 hpa. (**G**) Percentage of Runx2 +Bglap + and Runx2 +Bglap OBs subtypes. Statistical analysis displayed on each graph corresponds to Mann-Whitney test with Mean ± SD (n=79 (PBS) and 89 (2DG) cryosections). ns: not significant; ** $p<0.01$; *** $p<0.0001$. See *Figure 4—source data 1*.

The online version of this article includes the following source data for figure 4:

**Source data 1.** Spreadsheets detailing the results of the impaired osteoblast dedifferentiation after glycolysis inhibition with 2DG, specifically (**C**) the relative gene expression analysis, and the quantification of (**F**) the number of Runx2 +Bglap + and Runx2 +Bglap cells, and (**G**) the percentage of osteoblasts subtypes (Runx2 +Bglap + and Runx2 +Bglap-).

genes and in *bmp2b*, accompanied by a decrease in *cyp26b1* in 2DG-treated fins. Thus, suppression of glycolysis seems to maintain the mature OB differentiation profile and to prevent their dedifferentiation after amputation, resembling a pre-amputation scenario.

We then evaluated whether glycolysis is necessary for other aspects of OB dedifferentiation namely migration toward the stump and cell cycle re-entry. Surprisingly, by measuring the relative cell displacement of the OB population in the first segment below the amputation region after 24 hpa, we observed that 2DG administration had no effect on the ability of OB to migrate and reach the amputation zone (*Figure 5A–E*). However, cell cycle re-entry, examined through a EdU 3 hr pulse assay in *bglap*:EGFP transgenics, was strongly affected by 2DG treatment (*Figure 5F–G'*) contrasting with control caudal fins. We observed a significant reduction of total number of Runx2 +Bglap + EdU +pre OBs per area (*Figure 5H*) and percentage of Runx2 +Bglap + EdU +within the Runx2 +Bglap + population (*Figure 5I*). By quantifying the remaining pre-OB population Runx2 +Bglap-, we also observed a reduction in the number and relative percentage of Runx2 +Bglap-EdU+pre OBs upon 2DG treatment (*Figure 5F–G'* arrows, H,I). This indicates that blocking glycolytic machinery has a severe impact in the number of pre-OB that re-enter the cell cycle, reducing their proliferative capacity. This inhibition of cell cycle re-entry by 2DG was also observed in the blastema primordium mesenchyme (*Figure 5—figure supplement 1A-F,H*) and the overlying epidermal cap (*Figure 5—figure supplement 1A-F,G*). Moreover, the decline in the capacity of pre-OB to re-enter the cell cycle and to proliferate could be correlated with defects in the activity of pathways known to be indispensable for blastema proliferation, such as, Wnt, Insulin and Fgf signaling pathways (*Lee et al., 2005*; *Poss et al., 2000*; *Stoick-Cooper et al., 2007*; *Chablais and Jazwinska, 2010*; *Wehner and Weidinger, 2015*; *Shibata et al., 2016*). This is predicted based on our results showing a downregulation of *wnt10a*, *igf2b* and *fgf20a* in 2DG-treated caudal fins in comparison to controls (*Figure 5J*). We further complement and confirmed our results using 3PO, a partial glycolytic inhibitor, in *bglap*:EGFP transgenic fish. In accordance with previous results, 3PO led to a reduction in cell cycle re-entry of Bglap +pre OB, mesenchymal and epidermal cells, measured by PCNA immunofluorescence (*Figure 5—figure supplement 1I-O*).

To exclude any effect of glycolysis inhibition on cell survival that could interfere with our observations, we performed a TUNEL assay at 24 hpa in controls and in 2DG-treated *bglap*:EGFP transgenic zebrafish. Except for the mesenchyme compartment, which showed a slight increase in the number of TUNEL-positive cells, the epidermis and Runx2 +pre OBs showed no major alterations upon 2DG treatment (*Figure 5—figure supplement 2A-L*). This suggests that glycolysis may support cell survival in the mesenchymal compartment, but not in the epidermis or in the pre-OB pools. Thus, blocking glycolysis is sufficient to inhibit OB dedifferentiation and cell cycle re-entry, without affecting their capacity to migrate or survive.

Taken together, our results reveal that enhancing glycolysis promotes mature OB dedifferentiation into pre-OB and enabling pre-OB and other lineages to re-acquire proliferative capacity within the blastema. We also provide evidence that energy metabolism controls these aspects of OB response to injury through a glycolysis-dependent transcriptional regulation. Therefore, these results provide solid evidence that metabolic reprogramming toward glycolysis is a novel and powerful conductor of the cell fate changes and cell cycle re-entry preceding blastema assembly.

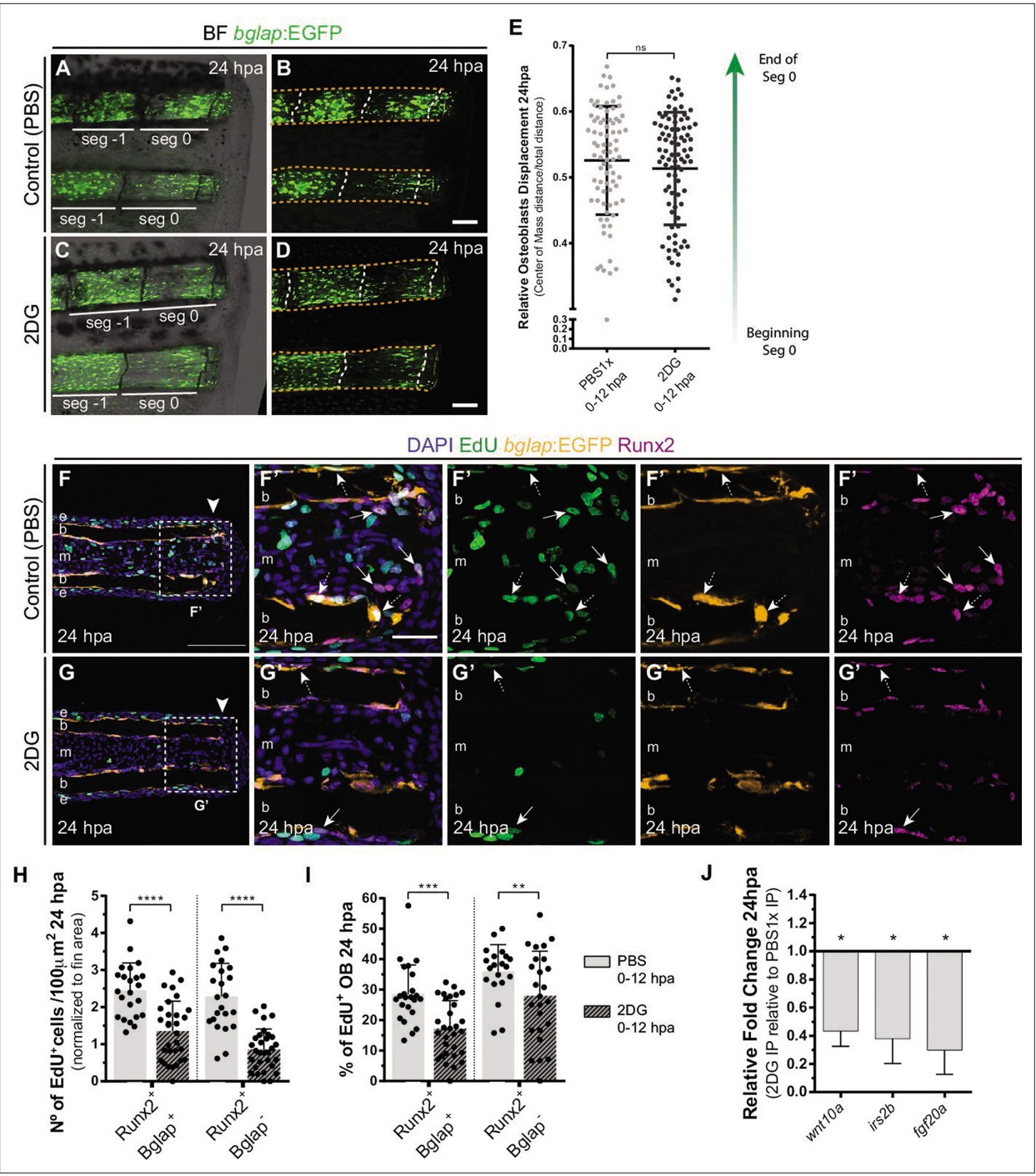

**Figure 5.** Inhibition of glycolysis impairs osteoblast cell cycle-entry. (**A–D**) Representative images of bglap:EGFP caudal fins at 24 hpa, treated with (**A–B**) vehicle (PBS) or (**C–D**) 2DG. Double white arrows indicate the anterior (**A**) and posterior (**P**) axis. White dashed lines indicate intersegment regions. Orange dashes lines delineate the bony-ray surface. (**E**) Measurement of relative OB displacement along segment 0, below the amputation plane, at 24 hpa in fins treated with vehicle (PBS) or 2DG. Statistical analysis on graph corresponds to Mann-Whitney test with Mean ± SD (PBS = 90, 2DG = 82 bony-rays). (**F-G'**) Representative cryosection images of 24 hpa bglap:EGFP (orange) caudal fins immunostained for Runx2 (magenta), labeled with EdU (green) and counterstained with DAPI (blue), in fish treated with (**F**) control (PBS) or (**G**) 2DG. Dashed boxes delineate amplified panels in F' and G'. Arrows indicate proliferative EdU +Runx2+Bglap cells. Dashed arrows indicate proliferative EdU +Runx2+Bglap + cells. Arrowhead indicates amputation plane. Scale bar represents 100 µm and 30 µm in amplified panels. (**H**) Total number of Runx2 +Bglap + and Runx2 +Bglap cells at 24hpa, in fins treated with vehicle (PBS) or 2DG. (**I**) Percentage of proliferative Runx2 +Bglap + and Runx2 +Bglap cells at 24hpa, in fins treated with vehicle (PBS) or 2DG. Statistical analysis displayed on each graph corresponds to Mann-Whitney test with Mean ± SD (n=23–30 cryosections). (**J**) Relative gene expression

*Figure 5 continued on next page*

*Figure 5 continued*

at 24 hpa in 2DG treated fins, compared to control. Statistical analysis with unpaired t test and Welch's correction (n=5 biological replicates). ns: not significant; *p<0.0001. See *Figure 5—source data 1*.

The online version of this article includes the following source data, source code, and figure supplement(s) for figure 5:

**Source code 1.** MatLab scripts to quantify the relative osteoblast displacement after caudal fin amputation in controls and after 2DG treatment.

**Source data 1.** Spreadsheets detailing the results of impaired osteoblasts cell cycle re-entry after glycolysis inhibition with 2DG, specifically the quantification of (**H**) the number of Runx2 +Bglap + and Runx2 +Bglap- EdU + cells, (**I**) the percentage of Runx2 +Bglap + and Runx2 +Bglap- EdU + cells and (**J**) the relative gene expression analysis.

**Figure supplement 1.** Inhibition of glycolysis prevents cell cycle re-entry.

**Figure supplement 1—source data 1.** Spreadsheets detailing the results of impaired cell cycle re-entry after glycolysis inhibition on individual fin tissues, specifically the quantification of the number of PCNA +and EdU + cells in the (**G**) epidermis and (**H**) mesenchyme after 2DG treatment, and (**I**) the number of PCNA + cells in the epidermis, mesenchyme, and osteoblasts after 3PO treatment.

**Figure supplement 2.** Inhibition of glycolysis has no effect on pre-osteoblasts cell death.

**Figure supplement 2—source data 1.** Spreadsheets detailing the results of the impact of glycolysis inhibition, after 2DG treatment, on pre-osteoblasts cell death, specifically the quantification of the number of TUNEL + cells in the (**I**) epidermis, (**J**) mesenchyme and (**K**) pre-osteoblasts, and (**L**) the percentage of TUNEL +osteoblasts.

## Glycolysis suppression leads to impaired blastema proliferation and defective distribution of new osteoblast subtypes

Based on our results so far, we demonstrated a fundamental role of glycolysis in governing OB dedifferentiation and the early stages of blastema formation. Subsequently, we aimed to investigate how prolonged inhibition of glycolysis would interfere with blastema organization and with de novo OB formation (*Figure 6A*). At 48 hpa, the blastema is subdivided in a patterning zone (PZ) and in a proximal (PB) and distal (DB) compartments. These regions are characterized by distinct OB subtypes, based on their maturation and proliferative state, exhibiting a proximal-distal hierarchical and overlapping distribution (*Wehner and Weidinger, 2015*; *Iovine, 2007*; *Nechiporuk and Keating, 2002*; *Poss et al., 2002*): proliferative Runx2[+] pre-OBs maintain the progenitor pool in the PB and, as they proliferate and become further away from the influence of the DB signals, they start to differentiate into proliferative lineage committed Runx2[+] Osx[+] immature OBs (*Stewart et al., 2014*; *Brown et al., 2009*), which will give rise to differentiated OBs in the PZ (*Figure 6B*; *Stewart et al., 2014*; *Brown et al., 2009*). To determine how glycolysis affects the general distribution of OB subtypes within the blastema, we exposed *runx2*:EGFP and *osx*:mCherry zebrafish to 2DG treatment through the first 48 hpa (*Figure 6A*). We observed that prolonged 2DG-treatment caused a severe abrogation of blastema organization (*Figure 6—figure supplement 1A-H*), with a strong reduction in *osx* (*Figure 6—figure supplement 1B,F,D,H*) and *runx2* (*Figure 6—figure supplement 1C,D,G,H*) when compared to control fins, indicating that 2DG administration strongly alters OB specific gene expression within the blastema. In normal regenerating conditions, immunofluorescence for Runx2 in *osx*:mCherry transgenics display a proper organization of the OB subtypes within the blastema (*Figure 6C–C'*). The cluster of Runx2 +pre OBs, referred as the Runx2 +Osx- subtype, is restricted to the PB and represents a smaller fraction of the OB lineage in the blastema (*Figure 6C, C' and E*), while immature OBs, referred as the Runx2 +Osx + subtype, reside in the interface between the PB region and the PZ, and correspond to the major OB subtype (*Figure 6C, C' and F*). In contrast, we observe that 2DG-treated fish had an accentuated decrease in the total number of Runx2[+]Osx[+] immature OBs (*Figure 6D, D' and F*), while Runx2[+]Osx[-] pre-OBs remain unchanged (*Figure 6D, D' and E*), leading to an imbalance between the OB populations within the blastema (*Figure 6G*). Similar results were obtained when using the glycolytic inhibitor 3PO (*Figure 5—figure supplement 1I-O*).

Afterwards, we decided to ascertain if the proliferative abilities of each OB subtype in control and 2DG-treated fish, through a EdU 3 h-pulse assay. As previously reported (*Stewart et al., 2014*), in control fins Runx2 +Osx + immature OBs exhibit a higher proliferation rate than Runx2 +Osx- pre-OBs (*Figure 6H, H', J and K*). Strikingly, glycolysis inhibition had a more profound impact on proliferation at 48 hpa (*Figure 6I–K*) than at 24 hpa (*Figure 5F–I*), with both OB subtypes exhibiting a significant reduction of their proliferative capacity (*Figure 6H–K*). This reduction was particularly noticeable in Runx2 +Osx + OB subtype. We also noticed that, as observed at 24 hpa (*Figure 5—figure supplement 1G,H*), the epidermis and mesenchyme were also affected, displaying a decrease in proliferation

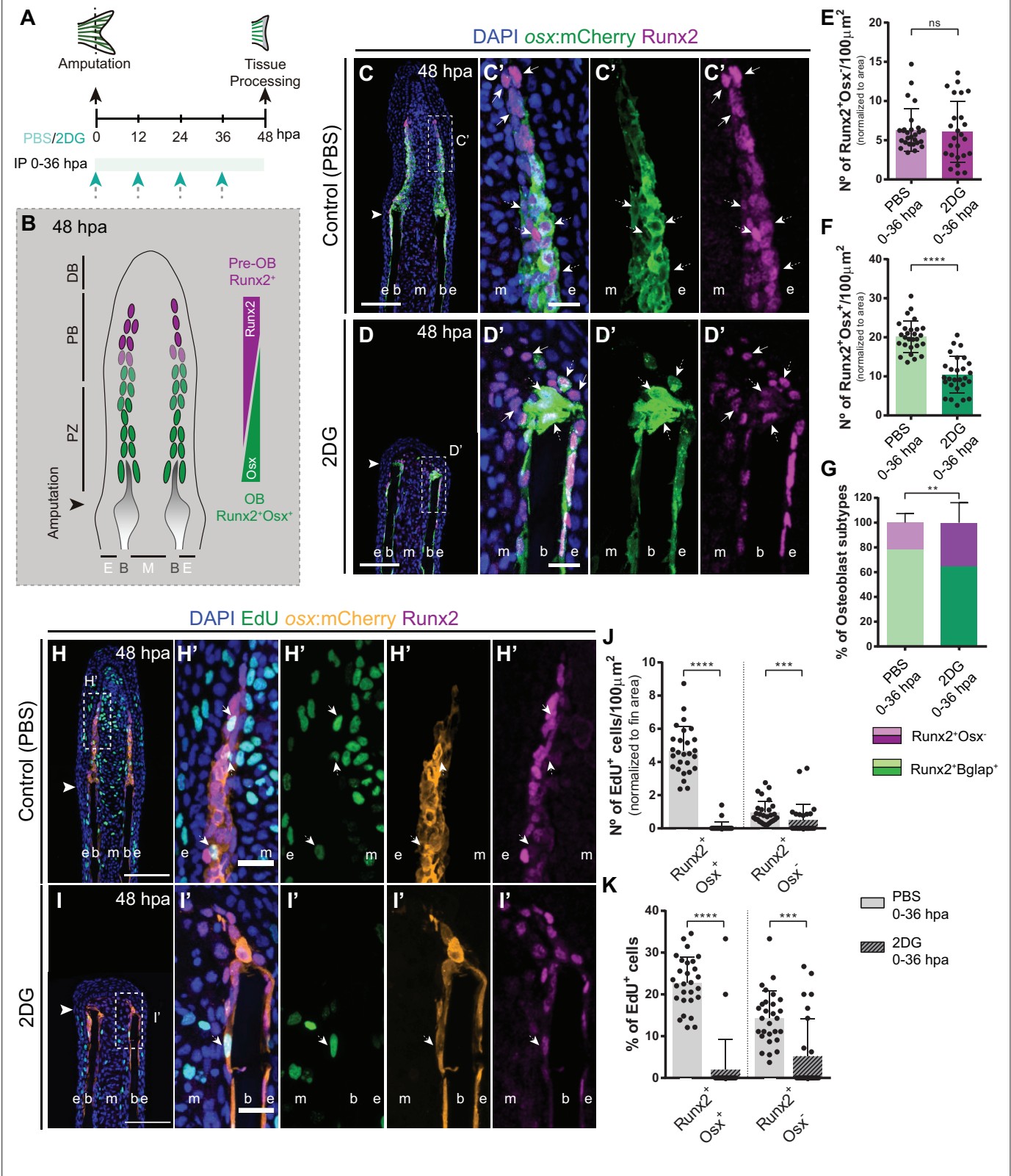

**Figure 6.** Inhibition of glycolysis affects formation of osteoblast subtypes and proliferation within the blastema. (**A**) Experimental design used to inhibit glycolysis. Fish are administered, via IP injection, with control (PBS) or 2DG every 12 hr, from fin amputation (0 hpa) until 48 hpa. (**B**) Schematic representation of the distribution of OBs subtypes along the blastema. (**C–D**) Representative cryosection images of 48 hpa osx:mCherry (green) caudal fins immunostained for Runx2 (magenta) and counterstained with DAPI (blue), in fish treated with (**C,C′**) PBS and (**D,D′**) 2DG. Dashed boxes represent

*Figure 6 continued on next page*

*Figure 6 continued*

magnified panels in C' and D'. Arrows indicate Runx2 +Osx- pre-OBs. Dashed arrows indicate Runx2 +Osx + immature OBs. (**E–F**) Total number of (**E**) Run2 +Osx and (**F**) Runx2 +Osx + subtypes in 48 hpa fins treated with PBS or 2DG (PBS = 27, 2DG = 25 cryosections). (**G**) Percentage of Runx2+/Osx- and Runx2 +Osx + subtypes in 48 hpa fins treated with PBS or 2DG. (**H-I'**) Representative cryosection images of 48 hpa osx:mCherry (orange) caudal fins immunostained for Runx2 (magenta), EdU (green) and counterstained with DAPI (blue), in fish treated with (**H,H'**) PBS and (**I,I'**) 2DG. Dashed boxes represent magnified panels in H' and I'. Arrows indicate proliferative Edu +Runx2+Osx- pre-OBs. Dashed arrows indicate proliferative Edu +Runx2+Osx + immature OBs. Arrowheads indicate amputation plane. Scale bar represents 100 μm and 20 μm in magnified panels. (**J**) Total number of Runx2 +Osx + and Runx2 +Osx- proliferative OBs subtypes at 48 hpa fins, treated with PBS or 2DG (PBS = 28, 2DG = 27 cryosections). (**K**) Percentage of proliferative Runx2 +Osx + and Runx2 +Osx OBs subtypes in 48 hpa caudal fins , treated with PBS or 2DG. E: epidermis; B: bone; M: mesenchyme. For all graphs, statistical analysis corresponds to Mann-Whitney and Mean ± SD are displayed. ns: not significant; **p<0.0001. See *Figure 6—source data 1*.

The online version of this article includes the following source data and figure supplement(s) for figure 6:

**Source data 1.** Spreadsheets detailing the results of the formation of osteoblast subtypes and proliferation within the blastema after glycolysis inhibition with 2DG, specifically the quantification of the number of (**E**) Runx2 +Osx- pre-osteoblasts and (**F**) Runx2 +Osx + osteoblasts, (**G**) the percentage of osteoblasts subtypes, (**K**) the total number of Runx2 +Osx + and Runx2 +Osx- EdU +osteoblast subtypes, and (**K**) the percentage of proliferative Runx2 +Osx + and Runx2 +Osx- osteoblast subtypes.

**Figure supplement 1.** Inhibition of glycolysis affects distribution of osteoblast subtypes in the blastema.

**Figure supplement 1—source data 1.** Spreadsheets detailing the results of the distribution of osteoblast subtypes in the blastema after glycolysis inhibition with 3PO, specifically the quantification of the number of (**K**) Runx2 +Osx and (**L**) Runx2 +Osx + cells.

**Figure supplement 2.** Inhibition of glycolysis impairs proliferation during blastema formation.

**Figure supplement 2—source data 1.** Spreadsheets detailing the results of cell proliferation after glycolysis inhibition with 2DG, specifically the quantification of the number of EdU +and PCNA + cells in the (**G**) epidermis and (**H**) mesenchyme.

**Figure supplement 3.** Mature osteoblasts accumulate at stump region after glycolysis inhibition.

**Figure supplement 3—source data 1.** Spreadsheets detailing the results of mature osteoblasts accumulation in the stump region after inhibition of glycolysis with 2DG, specifically (**C**) the quantification of the number of Runx2 +Bglap + cells.

**Figure supplement 4.** Inhibition of glycolysis impairs bone regeneration, but not fin and bony-ray integrity in uninjured conditions.

**Figure supplement 4—source data 1.** Spreadsheets detailing the results of inhibiting glycolysis with 2DG in uninjured caudal fins, specifically the quantification of (**L**) fin area to width ratio, and (**N**) the average of bony-ray length to width ratio.

in 2-DG-treated fish (*Figure 6—figure supplement 2A-H*). Given the results obtained, the most likely interpretation is that both pre-OB and immature OB populations accumulate at the stump region since they are unable to proceed in the cell cycle and divide. In fact, corroborating these observations, when monitoring Runx2 +Bglap + pre-OB, using *bglap*:EGFP line, at 48 hpa, we observed that while in normal regenerating conditions dedifferentiated OBs were found near the stump region, contributing to the blastema (*Figure 6—figure supplement 3A,A',C*) as previously demonstrated (*Knopf et al., 2011*), after 2DG inhibition these cells remained accumulated at the stump, indicative of their inability to progress during the regenerative process (*Figure 6—figure supplement 3B,B',C*). These data let us to further hypothesized that the Runx2 +Osx + immature OBs found near the stump region in the 2DG context (*Figure 6D, D', I, I'*), are, at least in part, still OBs that were unable to fully dedifferentiate after fin amputation. Also, it is noteworthy mentioning that after 2DG administration, Runx2 +Osx- pre-OBs are still able to be recruited at this stage, but do not increase in total numbers due to inability to proliferate. Interestingly, this may indicate that, although OB dedifferentiation is compromised after blocking the glycolytic influx, pre-OB may be generated by alternative sources. By extending 2DG treatment into the outgrowth and patterning phase of regeneration (*Figure 6—figure supplement 4A*), we observe that while control fish are able to efficiently reconstruct the lost skeletal tissue at 7 days post-amputation (dpa) (*Figure 6—figure supplement 4C,D*), in 2DG-treated animals bone regeneration was completely abolished (*Figure 6—figure supplement 4E, F*). Although we observe a clear effect of 2DG in inhibiting new bone regeneration, sustained 2DG exposure for 7 consecutive days (*Figure 6—figure supplement 4B*) seemed to have no impact on uninjured caudal fin morphology (*Figure 6—figure supplement 4G-J*). Uninjured fish subjected to 2DG possess similar caudal fin morphological parameters when compared to controls, namely the caudal fin area to width ratio (*Figure 6—figure supplement 4K,L*) and the bony-ray length to width ratio (*Figure 6—figure supplement 4M,N*), suggesting that during this protocol 2DG does not compromise caudal fin integrity.

Overall, these data demonstrate an indispensable role of glycolysis in regulating blastema proliferation and compartmentalization with important implications for new OB generation and bone formation.

## Discussion

### Metabolic adaptation as an early response to caudal fin injury

OB dedifferentiation has been suggested to occur at the end of wound healing phase (0–18 hpa) and during the blastema induction phase (12–24 hpa) (*Knopf et al., 2011*; *Sousa et al., 2011*; *Stewart et al., 2014*). Here, by providing a deeper characterization of OB dedifferentiation, we demonstrate that this process is triggered as early as 6 hpa, in parallel with the initial wound healing response (*Chen et al., 2016*). Moreover, our transcriptomic analysis of isolated OBs revealed a dynamic transcriptional response at 6 hpa in comparison to OBs from uninjured conditions. This provides the first molecular characterization of OBs preceding the dedifferentiation stage, highlighting that mature OBs start changing their transcriptome earlier than expected and that the first hours after amputation are crucial for the transcriptional and phenotypic alterations leading to dedifferentiation. The set of differentially expressed genes unveils potential new players worth revisiting in the future. Our study uncouples OB response from surrounding tissues, and addresses the early stages of fin regeneration, which are the least investigated. In fact, most published data focus on time points from 24 hpa onwards, when wound closure has finished, blastema formation is in progress and consequently initial cell identity transitions have been dictated, potentially missing initial regulators of dedifferentiation. Importantly, we show that at 6 hpa OB prioritise lactate-producing glycolysis, when compared to OB from uninjured fins. Additionally, we also observed a similar response at 6 hpa in the whole fin stump, corroborated by gene expression and metabolomic data. These alterations persist at least until 24 hpa, when the blastema primordium is being assembled. Moreover, we demonstrate that this change in metabolism to enhance glycolysis occurs concomitantly with an increase in mitochondria fission. We also show that glycolysis is indispensable to support blastema formation and regeneration. Blocking glycolysis leads to a complete blastema suppression, with a single injection of 2DG at 0 hpa being sufficient to induce aberrant blastema formation. These results indicate that OBs and other cell lineages respond to amputation by undergoing a change in the metabolic profile that favours glycolysis (*Figure 7*). Furthermore, glycolysis is necessary from the early onset of regeneration and the time interval when these changes in metabolism happen appears to be fundamental for the initiation of regeneration. It is possible that early wound response signals are important to trigger changes in metabolic signature. One potentially relevant event described at this stage is reactive oxygen species (ROS) production (*Gauron et al., 2013*). ROS and cellular metabolism are tightly connected as ROS are a by-product of mitochondrial oxidation (*Zorov et al., 2014*) and produced by epithelial cells upon damage (*Cordeiro and Jacinto, 2013*). ROS are shown to activate important molecules, such as HIF-1α, which has been shown to promote metabolic reprogramming toward glycolysis in other contexts (*Zhao et al., 2017*; *Nagao et al., 2019*; *Sies and Jones, 2020*). It would be interesting to evaluate whether ROS production is necessary to stimulate glycolysis during caudal fin regeneration.

### Metabolic adaptation in the caudal fin as a prelude to the stem cell state

Our results provide the first evidence that OB and other cell types respond to amputation by engaging metabolic routes that boost lactate-producing glycolysis instead of OXPHOS, thereby acquiring metabolic traits of stem cells. It is well described that embryonic and adult stem cells exhibit metabolic preferences distinct from their differentiated progeny. Both primed embryonic stem cells (*Folmes et al., 2012*; *Tsogtbaatar et al., 2020*; *Mathieu and Ruohola-Baker, 2017*; *Wanet et al., 2015*; *Prigione et al., 2015*) and quiescent and proliferating adult stem cells (*Ito and Suda, 2014*; *Ly et al., 2020*; *Wei et al., 2018*; *Intlekofer and Finley, 2019*; *Ito and Ito, 2016*) seem to rely on glycolysis, and once differentiated they undergo a metabolic rewiring to increase mitochondrial biogenesis and OXPHOS. This reflects an essential role of glycolysis in periods of rapid cellular growth, while oxidative metabolism is preferred in mature cells to maintain homeostasis (Lunt el al., 2011; *Gándara and Wappner, 2018*). Prioritizing glycolysis entails several advantages for rapid proliferating cells: fuels biosynthetic pathways necessary to sustain rapid cell growth and division by generating intermediaries

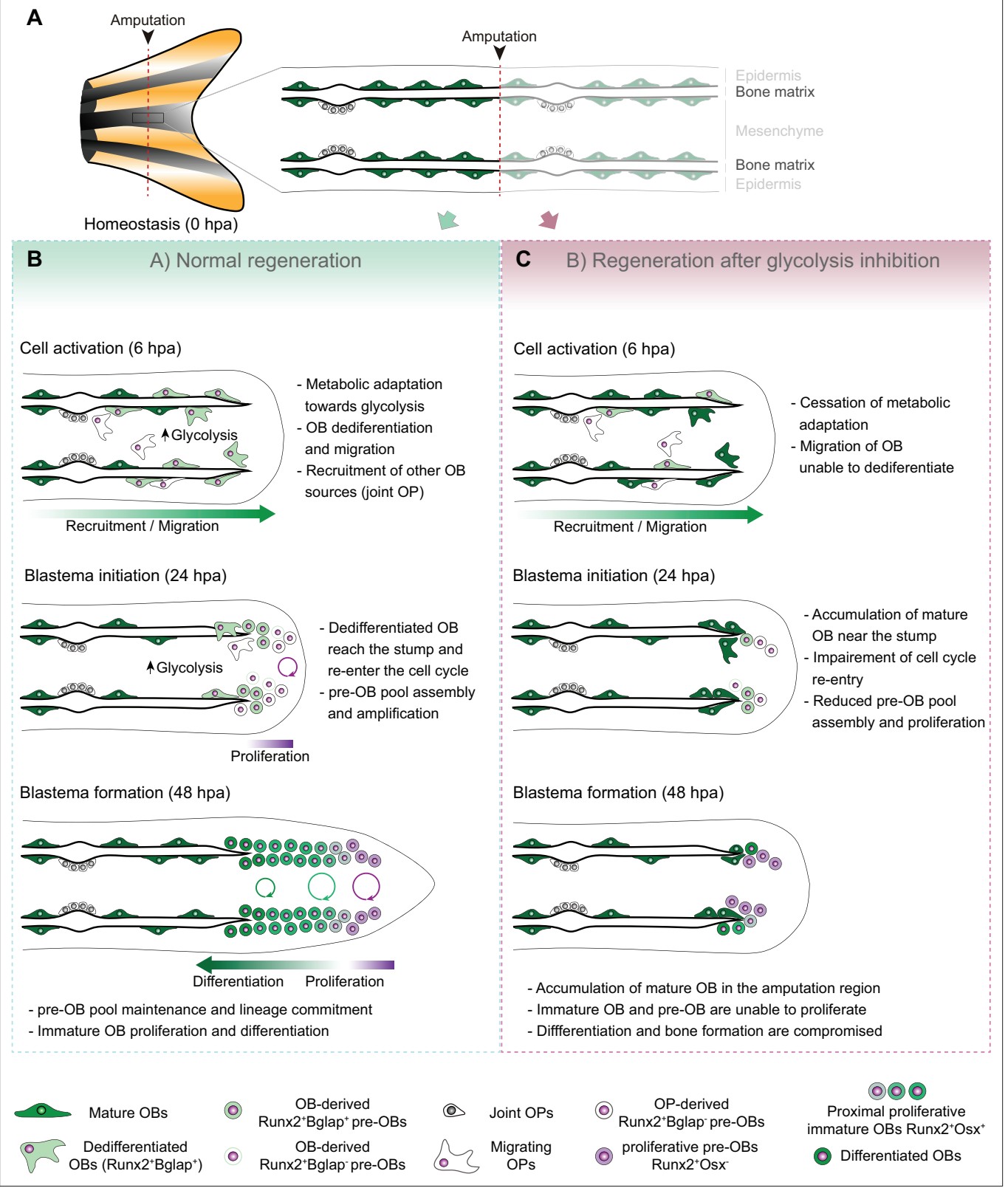

**Figure 7.** Model for the role of glucose metabolism during caudal fin regeneration. (**A**) In homeostasis, mature OBs reside in close contact with the bony-ray surface, secreting the collagenous bone matrix. (**B**) Upon caudal fin amputation, OBs and other cell types in the regenerating fin respond by undergoing a metabolic adaptation that stimulates glycolysis and is essential for regeneration to proceed. Enhancing glycolytic influx promotes OB dedifferentiation, by releasing Cyp26b1 from NF-ΚB repression, and cell cycle re-entry, by interfering with the master regulation of caudal fin

*Figure 7 continued on next page*

*Figure 7 continued*

proliferation Fgf20a, thereby enabling OBs to act as progenitor cells. Moreover, glycolysis is necessary to maintain the correct proliferative ability and distribution of OBs populations within the blastema, during its formation. (**C**) Glycolysis inhibition has a severe impact on OB dedifferentiation and pre-OBs pool assembly, which supports new OB formation and proliferation, ultimately leading to impaired bony-ray regeneration and suppression of blastema formation.

for macromolecules synthesis (e.g. nucleic acids, lipids; and non-essential amino acids); the rate ATP generation is faster through glycolysis than mitochondrial glucose oxidation (*DeBerardinis et al., 2008*; *Liberti and Locasale, 2016*; *Vander Heiden et al., 2009*; *Lunt and Vander Heiden, 2011*; *Yadav et al., 2020*); and finally, pathways branching from glycolysis also provide intermediate metabolites necessary for post-translational modification of proteins, histones and DNA (e.g. acetylation, methylation, phosphorylation, or glycosylation) (*Tarazona and Pourquié, 2020*; *Sun et al., 2022*; *Moussaieff et al., 2015*; *Etchegaray and Mostoslavsky, 2016*; *Ryall et al., 2015*). The latter, extends the connection between metabolism and modulation of intracellular signaling pathways, and the epigenome, to control gene expression programs that change cell function and fate (*Tarazona and Pourquié, 2020*; *Sun et al., 2022*; *Moussaieff et al., 2015*; *Etchegaray and Mostoslavsky, 2016*; *Ryall et al., 2015*). One of the best reported examples occurs during induced-pluripotent stem cell (iPSC) reprogramming, in which the switch toward a glycolytic metabolism happens before the expression of endogenous reprogramming factors (*Cliff and Dalton, 2017*; *Folmes et al., 2011*), implying that metabolic reprogramming is a cause rather that a consequence of cell reprogramming.

## Caudal fin metabolic adaptation resembles the Warburg effect

Metabolic reprogramming has also emerged in a disease context as a cancer hallmark and first described in cancer cells as the 'Warburg effect' (i.e. aerobic glycolysis) in which cancer cells use primarily glycolysis, resulting in lactate production, instead of pyruvate oxidation through OXPHOS (*DeBerardinis et al., 2008*; *Intlekofer and Finley, 2019*; *Warburg et al., 1927*; *Yadav et al., 2020*). The metabolic changes that occur during fin regeneration share several parallels between cancer metabolism pathophysiology, namely preference for glycolysis to support proliferation and elevated levels of glutamine, an essential nutrient that supplies cancer metabolism. Besides functioning as a precursor for nucleotides and amino acid synthesis, glutamine can be converted into glutamate, a metabolic intermediate with various fates in proliferating cells (e.g. protein synthesis, and incorporation into the TCA) (*Lu et al., 2010*; *Melissa, 2016*). Interestingly, like cancer cells, our data points to an important role of glutamine and glutamate for the assembly of the blastema primordium, as our metabolome studies show an increase by 100- and 400-fold in glutamine and glutamate at the beginning of blastema formation, respectively. Cancer cells also produce high levels of lactate where it is often regarded as an important oncometabolite (*Loeffler et al., 2018*), correlated with cancer-induced angiogenesis, invasion, metastasis, and immunosuppression (*de la Cruz-López et al., 2019*; *Parks et al., 2020*). Our studies show not only an increase in lactate during the initial stages of regeneration, but also reveal that inhibition of pyruvate conversion to lactate leads to defects in blastema formation, although milder when compared to glycolysis inhibition. This indicates that lactate production may also contribute to proper blastema formation. The mechanisms by which the glutamine and glutamate cycle and lactate influence blastema formation should be addressed in future studies. Unexpectedly, albeit aerobic glycolysis is known to support cancer cell migration, our results show that glycolysis is not required for mature OB recruitment and motility. It would be interesting to evaluate how alterations in glucose metabolism are regulated throughout the regenerative process, without falling into tumorigenesis.

## Metabolic adaptation is necessary for cell fate transitions and blastema proliferation

Our data shows that enhancing glycolysis serves to adapt to the cellular demands of regeneration, but it also seems to have the power to dictate several aspects of the regeneration program, including modulation of mature OB dedifferentiation. Previous studies have shown that OB dedifferentiation is a result of the activity of the NF-kB-RA axis (*Mishra et al., 2020*; *Blum and Begemann, 2015*). In homeostasis, NF-kB supports RA signalling, by blocking the expression of *cyp26b1*, the RA-degrading enzyme, maintaining OB differentiation. After amputation, NF-kB becomes inactivated and Cyp26b1

suppression is lifted, thereby protecting OB from RA and promoting their dedifferentiation (*Mishra et al., 2020*; *Blum and Begemann, 2015*). We show that blocking glycolysis leads to NF-kB signalling stimulation and decrease in *cyp26b1* expression, providing evidence that increase in glycolytic activity precedes and is necessary to induce mature OB reprogramming into pre-OB. In addition, our data shows that glycolysis is necessary to support pre-OB cell cycle re-entry and to sustain blastema proliferation (*Figure 7*) and can be linked to *fibroblast growth factor 20* a (*fgf20a*), which is fundamental for blastema initiation and proliferation during regeneration (*Poss et al., 2000*; *Shibata et al., 2016*; *Whitehead et al., 2005*). Mutants for *fgf20a* fail to form a functional blastema and are unable to proliferate (*Whitehead et al., 2005*). Accordingly, blocking Fgf receptor 1 activity leads to a similar phenotype (*Lee et al., 2005*; *Poss et al., 2000*), but does not impair OB dedifferentiation (*Knopf et al., 2011*), suggesting that its primary role is to regulate blastema proliferation. Our work suggests that glycolysis promotes not only the expression of *fgf20a*, but also of other ligands that cooperate to induce *fgf20a* expression, such as *igf2b* and *wnt10a* (*Stoick-Cooper et al., 2007*; *Chablais and Jazwinska, 2010*). Since these pathways are part of a general mechanism triggered upon amputation to stimulate proliferation, it is not surprising that glycolysis inhibition caused an overall reduction of proliferation. In addition, the presented data indicates that glycolysis is necessary until the end of blastema formation, to generate new OBs and to maintain a proper balance between OB subtypes within the blastema (*Figure 7*). Decline in the total number of immature OB and in the proliferative rates of distal pre-OB and of proximal immature OB populations observed upon glycolysis inhibition, can be accounted, at least in part, by the pronounced effects of glycolysis inhibition on blastema proliferation. Importantly, these results corroborate the idea that regeneration benefits from glycolysis both in terms of biomass generation, to support cell proliferation, and by inducing the expression of powerful mitogens, like *fgf20a*. Noteworthy, besides mature OBs, pre-OB can also derive from a population of OB progenitor that resides in the joint regions of the fin (*Ando et al., 2017*). Thus, mature OB and joint-associated progenitors may act as complementary sources that supply the pre-OB pool. In fact, we observed that glycolysis inhibition leads to a diminished number of pre-OBs before blastema formation, yet this number is back to normal after blastema formation. Since blocking glycolysis prevented OB dedifferentiation, we could speculate that over time OB progenitors from the joints were able to replenish the pre-OB pool. Further work is needed to test this hypothesis and the impact of glucose metabolism in supporting OP activation and contribution for new OB formation. In general terms, this study provides the first line of evidence that metabolic adaptation towards glycolysis governs mature OB dedifferentiation and blastema proliferation. To some extent, this is mediated through glycolysis-driven changes in gene expression that allows to uncouple dedifferentiation from acquisition of proliferative capacity.

## Metabolic adaptation as a conserved mechanism in regenerative contexts

Regeneration is in its essence an anabolic process. After an insult, reconfiguration of the extracellular milieu can induce metabolic adaptations that are fundamental to accommodate new cellular functions that support growth and cell fate decisions necessary for regeneration. In line with our data, other animals with enhanced regenerative abilities, such as planarians (*Osuma et al., 2018*) or amphibians (*Alibardi, 2014*; *Varela-Rodríguez et al., 2020*) also show a predominance of glycolysis upon injury to sustain proliferation. This indicates that metabolic rewiring towards glycolysis might be a conserved mechanism necessary for the regenerative process. Importantly, changes in glucose metabolism were also shown to be necessary for regeneration of other zebrafish tissues. Like OBs in the fin, after cardiac injury, regeneration is achieved via dedifferentiation and proliferation of cardiomyocytes near the injury (*Kikuchi et al., 2010*; *Jopling et al., 2010*). Recent studies have demonstrated that these cells switch to a glycolytic metabolism necessary for their dedifferentiation and proliferation (*Honkoop et al., 2019*; *Fukuda et al., 2020*). Moreover, regeneration of the embryonic tail was shown to rely on glycolysis to support blastema formation (*Sinclair et al., 2021*). Glycolysis was required to fuel the hexosamine pathway (*Sinclair et al., 2021*), which is responsible for glycosylation of proteins associated with cell signaling, gene transcription, and EMT (*Akella et al., 2019*; *Reily et al., 2019*). Given these results and the similarities between larval tail and adult caudal fin regeneration, it would be important to examine the function of hexosamine pathway during fin regeneration. In contrast to zebrafish, mammals possess poor capacity to perform epimorphic regeneration of complex structures,

with only a few examples, such as amputated ear and digit tips (*Seifert and Muneoka, 2018*; *Johnson et al., 2020*). In mice models of ear and digit injuries, regeneration is impaired by OXPHOS inhibition, suggesting that in this context OXPHOS is required to mediate regeneration (*Shyh-Chang et al., 2013*). In contrast, the MRL mice strain, which has an enhanced regenerative capacity in comparison to other mice, showed an increase of aerobic glycolysis over OXPHOS after injury of several organs (*Naviaux et al., 2009*; *Heber-Katz, 2017*). This indicates that further studies are necessary to clarify the potential role of glucose metabolism during mammalian regeneration. Regarding bone, disruption of the metabolic profile of OBs and OB sources (e.g. mesenchymal stem cells) might also have important implications for bone repair after injury and in certain pathological conditions (e.g. osteoporosis), as they influence OB identity status and function (*Lee et al., 2017*; *Loeffler et al., 2018*; *van Gastel and Carmeliet, 2021*; *Karner and Long, 2018*). Cell metabolism can potentially be a target in the contexts of fracture healing or bone diseases, to stimulate the repair process, or to prevent OB dysfunction.

## Concluding remarks

The data described here provides clear evidence that a metabolic reprogramming favouring anaerobic glycolysis occur at early stages of adult regeneration and are an integral component of the regenerative program. This is in accordance with recent regeneration studies performed in other systems and resembles many traits of the Warburg effect observed in cancer cells. Our data indicates that OB and possibly other cell lineages favour glycolysis, to engage a specialized genetic program that enables them to act as progenitor cells. We unveil a novel and fundamental role of glycolysis in mediating mature OB dedifferentiation and cell cycle re-entry and supporting blastema assembly and proliferation. Moreover, we have uncoupled the effects of glycolysis in mediating OB dedifferentiation from proliferation by identifying distinct downstream transcriptional targets of the glycolytic metabolism. This provides evidence that the role of glucose metabolism in regeneration is not limited to sustain macromolecule synthesis and energy production. Overall, our findings support a notion that glucose metabolism has a powerful instructive role in regulating lineage-specific programs and generic responses to injury that induce changes in cell identity and function, crucial to prompt bone regeneration.

## Materials and methods
### Zebrafish lines maintenance and caudal fin amputation

Wild-type (WT) AB and transgenic zebrafish lines, namely *Tg(osterix:mCherryNTRo)$^{pd4}$* (*Singh et al., 2012*) (referred as *osx*:mCherry), kindly provided by Kenneth Poss, *Tg(ola.Bglap:EGFP)$^{hu4008}$* (referred as *bglap*:EGFP) and *Tg(Has.RUNX2-Mmu.Fos:EGFP)$^{zf259}$* (*Knopf et al., 2011*) (referred as *runx2*:EGFP), kindly provided by Gilbert Weidinger, and *Tg(Xla.Eef1a1:mlsEGFP)* (referred as MLS-GFP), kindly provided by Seok-Yong Choi (*Kim et al., 2008*) were maintained in a circulating system with 14 hr/day and 10 hr/night cycle at 28 °C (*Westerfield, 2000*). All regeneration experiments were performed in 4–18 months-old fish and transgenics used as heterozygotes. Caudal fin amputations were performed in fish anesthetized with buffered 160 mg/mL MS-222 (Sigma, E10521), using a sterile scalpel to remove approximately one half of the fin, as previously described (*Poss et al., 2000*). Fish were left to regenerate in an incubator at 33°C ± 1°C with water from the circulating system until defined time-points. Regenerated fins were collected from anaesthetized fish, and either processed for cryosectioning, stored in Trizol for RNA isolation, handled for Mass-spectrometry (MS) or for flow cytometry.

### Pharmacological and chemical treatments

For pharmacological treatments via intraperitoneal injections (IP), fish were randomized and subjected to IP injections at the designated time-points, with either 2DG (Sigma-Aldrich, 0.5 mg/g diluted in 1 x Phosphate Buffered Saline (PBS)), S.O. (Sigma-Aldrich, 0.6 mg/g diluted in 1 x PBS), UK-5099 (Sigma-Aldrich, 0.02 mg/g, diluted in a mixture of 1 x PBS and DMSO (1:1)) or with corresponding vehicle (Control). IP injections were performed with an insulin syringe U-100 G 0.3 mL and a 30 G needle (BD Micro-fine) inserted close to the pelvic girdle. For 3PO (Sigma-Aldrich) and MB-6 (Calbiochem) treatments, compounds were diluted in DMSO and added to water from the circulating system to a

final concentration of 15 µM and 2.5 µM, respectively, and controls with equivalent amount of vehicle. For all experiments water was replaced daily and fish left to regenerate until the desired time-point.

For glucose uptake assay, fish were administered with the glucose analogue 2-NBDG (Sigma-Aldrich, 25 µmol/kg, from stock solution dissolved in DMSO) via IP injection 1 hr prior to imaging. For S-phase labeling, fish were subjected to caudal fin amputation and administrated with Ethynyl-2´-deoxyuridine (EdU, Thermo Scientific: C10337, 20 µL of 10 mM solution diluted in 1 x PBS) via IP injection 3 hr prior to caudal fin collection.

## Total RNA isolation and quantitative real-time PCR (qPCR)

For gene expression analysis, caudal fin composed of the regenerated tissue and one bony-ray segment proximal to the amputation plane were collected. Pools from 4 to 5 caudal fins were used per biological replicate. Briefly, samples were homogenized in Trizol reagent (Invitrogen, 15596026) for cell disruption and RNA extracted as previously described (*Brandão et al., 2019*), using the RNeasy Micro kit (Qiagen, 74004) according to manufacturer's protocol. cDNA was synthesized from 1 µg total RNA for each sample using the Transcriptor High Fidelity cDNA Synthesis Kit (Roche, 05081963001), with a mixture of oligo dT and random primers. All qPCR primers are listed in *Supplementary file 2a*. qPCR was performed using a FastStart Essential DNA Green Master Mix (Roche, 4385617) and a Roche LightCycler 480. Cycle conditions were: 15 min pre-incubation at 95 °C and three-step amplification cycles (45 x), each cycle for 30 s at 95 °C, 15 s at 60 °C, and for 30 s at 72 °C.

## Skeletal staining and immunofluorescence in cryosections

In vivo Alizarin red S (ARS, Sigma-Aldrich) staining in the *bglap*:EGFP transgenic was performed prior to caudal fin amputation as previously described (*Bensimon-Brito et al., 2016*). Briefly, fish were incubated in a 0.01% ARS solution, dissolved in water from the circulating system and pH adjusted to 7.4 with a KOH solution, for 15 min in the dark and rinsed three times, 5 min each. Caudal fins were amputated and imaged at specific time-points post-amputation.

For calcein staining in WT AB strain, caudal fins were collected at predefined time-points post-amputation and post-treatment. Fins were fixed overnight (ON) in 4% paraformaldehyde (in 1 x PBS). Fins were washed in 1 x PBS and immersed into a 0.2% calcein solution (2 g of calcein powder (Sigma-Aldrich, C0875-56) in 1 L of 1 x PBS, pH 7.4) for 15 min. Afterwards, fins were washed five times in 1 x PBS, 10 min each, and left for 10 min in 1 x PBS to allow the unbound calcein to diffuse out of the tissues (*Brandão et al., 2019*; *Du et al., 2001*) and imaged.

Tissue processing for cryosections was performed as previously described (*Brandão et al., 2019*). Shortly, fins were collected, fixed overnight (ON) in 4% paraformaldehyde (in 1 x PBS) and stored in 100% methanol (MeOH) at –20 °C, until subsequent analysis. They were then gradually rehydrated in a series of MeOH/1 x PBS (75%, 50%, and 25%) and incubated ON in 30% sucrose (Sigma-Aldrich, diluted in 1 x PBS). For EdU labeling, caudal fins were fixed and directly incubated in 30% sucrose solution. Fins were then embedded in 7.5% gelatin (Sigma-Aldrich)/ 15% sucrose in 1 x PBS and subsequently frozen in isopentane at –70 °C and stored at –80 °C. Longitudinal caudal fins sections were obtained at 12 µm using a Microm cryostat (Cryostat Leica CM3050 S) and slides stored at –20 °C. For immunofluorescence on cryosections, slides were thawed for 15 min at room temperature (RT), washed twice in 1 x PBS at 37 °C for 10 min and subjected to an antigen retrieval step, which consisted of a 15 min incubation at 95 °C with sodium citrate buffer (10 mM Tri-sodium citrate with 0.05% Tween20, pH 6). Slides were then incubated in 0.1 M glycine (Sigma-Aldrich, in 1 x PBS) for 10 min, permeabilized in acetone for 7 min at –20 °C and incubated for 20 min in 0.2% PBST (1 x PBS with 0.2% Triton X-100). At this point, cryosections used for EdU labelling were incubated with the labelling solution according to the manufacturer's protocol (Thermo Scientific: C10337). For TUNEL labelling assay, cryosections were permeabilized in a sodium citrate solution (0.1% sodium citrate and 0.1% Triton X-100 in 1 x PBS) and labeled according to the manufacturer's protocol (Roche, 11684795910). Afterwards, they were incubated in a blocking solution of 10% non-fat dry milk in PBST for 2–4 hr at RT. Slides were then incubated with primary antibodies diluted in blocking solution, ON at 4 °C (for antibody details see *Supplementary file 2b*). On the following day, slides were washed with PBST 6 times, 10 min each, and incubated with secondary antibodies (*Supplementary file 2c*) diluted in blocking solution, for 2 hr at RT and protected from light. Subsequently, slides were washed three times, 10 min each, in PBST and then counterstained with 4',6-diamidino-2-phenylindole (DAPI;

0.001 mg/mL in 1 x PBS, Sigma-Aldrich) for 5 min in the dark, for nuclei staining. Slides were then washed three times with PBST, 10 min each, mounted with fluorescent Mounting Medium (DAKO) and stored at 4 °C protected from light until image acquisition.

## Flow cytometry

For fluorescence-activated cell sorting (FACS) of OB, caudal fins from *bglap*:EGFP transgenic line were amputated, tissue collected at specific time-points during regeneration and dissociated into single-cell suspensions. For that, fins were incubated for 20 min at 28 °C with vigorous shaking in a solution of Liberase DH Research Grade (0,05 mg/ml in 1 x PBS, Roche). Cell suspensions were passed through a 30 µm filter (CellTricks, Sysmex) and centrifuged at 300 g for 5 min at 4 °C. Cell pellets were resuspended in 1 x PBS with 10% fetal bovine serum (Biowest). FACS was carried out on a MoFlo high-speed cell sorter (Beckman Coulter, Fort Collins, USA) using a 488 nm laser (200 mW air-cooled Sapphire, Coherent) at 140 mW for scatter and a 530/40 nm bandpass filter for GFP measurements. Cell debris and aggregates were removed from the analysis. The fluorescence scatter (Comp-FL Log::GFP) was used to separate cells according to their GFP fluorescence intensity with a maximum of stringency to avoid cross-contamination. Zebrafish WT AB strain was used as a negative control to set the GFP-positive population. The instrument was run at a constant pressure of 207 kPa (30 psi) with a 100 µm nozzle and frequency of drop formation of approximately 40 kHz. Two and three independent biological replicates were performed for each condition at 0 and 6 hpa respectively. For each, 300 GFP-positive cells were collected directly into lysis and RNA stabilization buffer (provided by OakLabs GmbH) and vigorously shaken for 1 min. To verify the quality of the samples, cell death and purity were measured. Cell death was measured by incubating the samples with propidium iodide (PI, Sigma-Aldrich), to a concentration of 1 µg/ml, and using the 488 nm laser for PI excitation and measure on the PI channel (613/20 BP). Only samples with cell death below 10–20% and purity above 90% were used for subsequent analysis. Samples were maintained at –80 °C until sent to OakLabs GmbH (Henningsdorf, Germany) for cDNA generation, microarray chip set up and data analysis.

## Osteoblast ArrayXS

To compare the transcriptome profiles of mature OB in homeostasis to OB during dedifferentiation, a genome-wide gene expression profiling was set up using the 8x60 K ArrayXS Zebrafish platform by Agilent and performed by OakLabs GmbH (Henningsdorf, Germany). The 8x60 K ArrayXS Zebrafish represents approximately a total of around 60,000 zebrafish transcripts, which includes 48,000 coding genes, 8075 non-coding genes and 19,140 predicted genes annotated in the Zv9 release 75. RNA quality was processed by Oaklabs using the 2100 Bioanalyzer (Agilent Technologies), the RNA 6000 Pico Kit and a photometrical measurement with the Nanodrop 2000 spectrophotometer (Thermo Fisher Scientific). Sample quality was evaluated based on the Bioanalyzer's RNA integrity number (RIN). Only samples with RIN ≥8 were used. Subsequently, 2 µL of the lysis and RNA stabilization buffer, from three biological replicates of each condition (0 and 6 hpa OBs), was used for cDNA synthesis and pre-amplification using the Ovation One Direct system (NuGEN). The generated cDNA was labeled with Cy3dCTP using the SureTag DNA Labelling Kit (Agilent) prior to microarray hybridisation. Microarray blocking, hybridisation and wash were performed using Agilent's Oligo aCGH/ChIP-on-Chip Hybridisation Kit, following the manufacturer's protocol. Ultimately, fluorescence signals were detected by the SureScan Microarray Scanner (Agilent Technologies), at a resolution of 3 µm for SurePrint G3 Gene Expression Microarrays and 5 µm for HD Microarray formats. This resulted in a raw data output of one-color hybridization using the Agilent's Feature Extraction software version 11. Raw data was then subjected to processing and analysis. Briefly, background signals were subtracted and then normalized using the ranked mean quantiles (*Bolstad et al., 2003*). For data quality control and to identify potential outlier samples, hierarchical clustering and a principal component analysis were performed. The retrieved data was used to compare the expression profiles of OB from 6 hpa with 0 hpa. Differential expression was tested using Welch's t-test with p-values adjusted according to the adaptive Benjamini-Hochberg procedure (*Benjamini and Hochberg, 2000*). Significantly differentially expressed genes were identified whenever p-value was lower than the 0.05 threshold, and log2 fold change < –1 or >1. Transcriptome datasets analyzed on this study were submitted to NCBI Gene Expression Omnibus archive with an accession number GSE194385.

## Liquid chromatography-mass spectrometry (LC-MS) analysis

For metabolite analysis, caudal fins were collected and snap-freeze in liquid nitrogen for 5 min and diluted in a mixture containing MeOH:dH2O (2:1) and an internal standard α-Aminobutyric acid (AABA, 2 mM final concentration). Samples were homogenized using tissue grinder for 5 s and using the ultrasound bath for 30 min at 4°. This was followed by sample centrifugation for 10 min at top speed at 4°, supernatant collected and stored at –20° (short storage) or –80 °C (long storage). Samples and internal standards were analyzed in a Dionex UltiMate 3000 UHPLC (Ultra-High Performance Liquid Chromatography) system coupled to a heated electrospray QExactive Focus mass spectrometer (Thermo Fisher Scientific, MA, USA). Three separate LC-MS assays were applied. For glutamine (Duchefa) and glutamate (Thermo Fisher Scientific) detection, we used an Acquity UPLC BEH Amide column and spectra were acquired in positive ionization mode using a method which consisted of several cycles of FullMS scans (75–1125 m/z; Resolution = 70,000 FWHM at 200 m/z). Glucose (Sigma Aldrich) and lactate (Alfa Aesar) detection were performed with acquisition in negative ionization mode. For every assay, four biological replicates (10 fins used per replicate) were used per condition and sample injection was performed in triplicate and a volume of 5 µL was applied.

## Image acquisition and processing

For regenerated area measurements, images of live anesthetised WT and transgenic adult caudal fins were acquired in a Zeiss Lumar V-12 fluorescence stereoscope equipped with a Zeiss axiocam MRc camera using a 0.8 X air objective (at 14 x zoom) controlled by Zen 2 PRO blue software. Images were acquired using transmitted light and the GFP (FS05) and/or TexasRed (FS45) filters, according to the fluorescent reporter expressed. Images were assembled using the Fiji software (*Schindelin et al., 2012*).

For 2-NBDG labeled WT caudal fins, images were acquired using in a Zeiss Axio Observer z1 inverted microscope for transmitted light and epifluorescence, equipped with an axiocam 506 monochromatic camera, using an EC Plan-Neofluar 5x0.16 NA air objective controlled by Zen 3 blue software. An image mosaic was acquired using transmitted light and the GFP (38HE) filter. Serial sections were acquired every 5 µms. For image processing, composite maximum intensity images and concatenation of several images along the caudal fin proximal-distal axis was performed using the Zen 3 blue software and images assembled using Fiji software (*Schindelin et al., 2012*).

For live-imaging analysis of OB migratory dynamics in vivo, *bglap*:EGFP transgenic fish were anesthetised and maintained in glass bottom Petri dishes. Imaging was performed in a confocal microscope Zeiss LSM 710 using the software ZEN 2010B SP1. Fish were imaged with a Plan-Neofluar 10x0.3 NA air objective using the 488 nm (emission windows:490–530 nm) and 568 nm (emission windows:570–650 nm), if counterstained with ARS, excitation wavelengths coupled with transmitted light PMT. Serial sections were acquired every 5 µms. For OB motility assay, time-lapse images were acquired always in the same region of the fin, capturing the first 2 segments below the amputation plane (segment 0 and segment –1) and the blastema region, and images acquired every 5 hr following amputation, during the first 25 hpa. For assessment of OB migration in vehicle and 2DG treated fish, time-lapse images were acquired at 0 and 24 hpa. For image processing, composite maximum intensity z-stack projections were made using the Fiji software (*Schindelin et al., 2012*). Time-lapses were assembled and computationally registered with the Fiji StackReg and MultiStackReg plugins (*Schindelin et al., 2012*).

Immunolabeled cryosections were analyzed in confocal microscopes Zeiss LSM 710 and Zeiss LSM 980 controlled by ZEN 2010B SP1 or ZEN 3.3, respectively. Cryosection images were acquired using a C-Apochromat 40x1.2 NA water objective with 0.6 x zoom, a step size of 1 µm, and 405, 488, 568, and 633 nm excitation wavelengths coupled with transmitted light PMT. Sequential images were acquired to capture the first segment below the amputation plane and the entire regenerated region. For image analysis and processing, composite maximum intensity z-stack projections were made using the Fiji software (*Schindelin et al., 2012*). When required, concatenation of several images along the proximal-distal axis of the same longitudinal section was performed using the Fiji plugin 3D Pairwise Stitching (*Schindelin et al., 2012*). To count and measure the number of mitochondria per cell in longitudinal cryosections of individual regenerating bony-rays, we used the surface tool from IMARIS software using a mitochondria surface detail of 0.1 µm and a split touch size of 0.5 µm.

For all cryosections of manipulated fish and corresponding control and time-lapse assays, images were acquired employing identical settings (magnification, contrast, gain and exposure time) and in identical/comparable regions. All Images were then processed using the Adobe Photoshop CS5 and Adobe Illustrator CC.

## Quantifications and statistical analysis

For qPCR analysis, all samples were analyzed in four to eight biological pools. For each biological pool, qPCR was performed for each target gene in three technical replicates. Gene expression values were normalized using the *elongation factor 1α* (*ef1α*, NM_131263) housekeeping gene and relative expression was calculated using the 2(-ΔΔC(T)) method (*Livak and Schmittgen, 2001*; *Figure 1—source data 1*).

Measurements of total regenerated area were obtained by delineating the fin regenerated area using the Area tool in Fiji. The regenerated area was then normalized to the corresponding total caudal fin width to avoid discrepancies related to the animal size, resulting in one measurement per animal. The percentage of regenerated area from caudal fins subjected to chemical treatments was calculated considering the average number of regenerated area values for the control condition as our 100%.

Total number of pre-OB (Runx2$^+$) and OB subtypes (Runx2$^+$Bglap$^+$ or Runx2$^+$Osx$^+$) in longitudinal cryosections was quantified by analyzing the number of labeled cells in relation to the imaged cryosection area (per 100 µm$^2$), determined using the Area tool on Fiji. Percentage of pre-OB and OB subsets was quantified by analyzing the number of cells in the imaged cryosection area in relation to the total number of OB lineage cells (Runx2$^+$ and Runx2$^+$Bglap$^+$or Runx2$^+$ and Runx2$^+$Osx$^+$).

Percentage of proliferating Bglap$^+$ OBs was quantified by analyzing the number of Bglap$^+$ PCNA$^+$OBs in the imaged cryosection area in relation to the total number of Bglap$^+$ OBs, per time-point analyzed. Total number of EdU$^+$ pre-OB (Runx2$^+$) and OB subtypes (Runx2$^+$Bglap$^+$ or Runx2$^+$Osx$^+$) was quantified by analyzing the number of labeled cells in relation to the imaged cryosection area (per 100 µm$^2$). Percentage of EdU$^+$ pre-OB (Runx2$^+$) and OB subtypes (Runx2$^+$Bglap$^+$ or Runx2$^+$Osx$^+$) was quantified by analyzing the number of EdU$^+$ cells within the cryosection area in relation to the total number of OB lineage cells (Runx2$^+$ and Runx2$^+$Bglap$^+$or Runx2$^+$ and Runx2$^+$Osx$^+$).

Total number of TUNEL$^+$ and EdU$^+$ cells was assessed by quantifying the number of labeled cells within each fin compartment (mesenchyme and epidermis), in relation to the corresponding compartment area (per 100 µm$^2$), determined using the Area tool on Fiji.

All quantifications were done using the Cell-counter plugin on Fiji in individual cryosections representing at least three different blastemas per animal and three to five animals per condition.

For quantification of OB motility during regeneration, live-imaging time-lapses of bony-rays, including the segment 0 and segment –1, were used. Quantification was performed using a custom Matlab script that performs all the workflow. Both GFP and brightfield (BF) channels were gaussian filtered (sigma 2) to reduce noise. Intersegment regions were found to define the boundaries between the segments analyzed using the BF channel and a sobel vertical algorithm, dilated with a vertical kernel and small connected components (200pixels) removed resulting in an average line profile for each bony-ray. Intersegments (peaks) were found using findpeaks matlab function. OB location was tracked by finding the global GFP intensity center (center of mass) in segment 0 and –1. GFP line profiles were calculated and summed in height and the intensity center of mass was found in each segment analyzed. The final result is expressed as a ratio (relative OB displacement) between the center of mass location and the total segment length (0: Anterior bias; 1: Posterior bias).

Number of mitochondria per cell was assessed by quantifying the number of mitochondria and the number of nuclei within the fin, using the Surface tool on IMARIS. The percentage of mitochondria volume was determined with the same tool, and, for each condition, the volumes of detected mitochondria were grouped into four distinct intervals: smaller than 0.1 µm$^3$, 0.1–1 µm$^3$, 1–10 µm$^3$ and larger than 10 µm$^3$.

For LC-MS, raw data was analysed using Xcalibur's Quan Browser (version 4.1.31.9, Thermo Scientific). The peaks corresponding to each compound of interest were identified by comparison with standards analysed in the same conditions. A mass tolerance of 5 ppm and a retention time window tolerance of 10 s were used. Peak areas used for relative quantitation were obtained using the Genesis method. Peak detection considered the nearest peak to the retention time defined for

each compound, with a minimum peak height (S/N) of 10. The peak area from AABA was used as an internal quantitation calibrant for the final quantitative data. Quantitation variability was assessed by the calculation of the relative coefficient of variance (CV %).

Statistical analysis was performed in GraphPad Prism v7 and statistical significance considered for $p < 0.05$. Statistical tests, p values, mean and error bars are indicated in the respective figure legends. For sample size see *Supplementary file 2d*. For OB ArrayXS, fold change was determined based on the normalised data set and expression ratios obtained. In this data set, a logarithmic base 2 transformation was performed (i.e. log2 (expression ratio)) to make the mapping space symmetric and the up-regulation and down-regulation comparable, prior to the significance test. The mean and standard deviations of the two sets of isolated OB samples (0 hpa and 6 hpa) were then compared using a Welch's t-test (or unequal variances t-test), generating 1 data set with the differential expressed genes between both conditions. Only p-values less than 0.05 were considered statistically significant and log2 values that lie between –1 and 1 were ignored.

## Acknowledgements

We are grateful to Lara Carvalho for reading the manuscript. Ana Teresa Tavares and Carolina Crespo for useful discussion and the Tissue Repair and inflammation group for support in many ways. Most of the work was funded by Fundacão para a Ciência e a Tecnologia by the following grants: PTDC/BIA-BID/29709/2017 and PTDC/BTM-SAL/29377/2017 to ASB, JB, RL and AJ; iNOVA4Health – UIDB/04462/2020 and UIDP/04462/2020; Associated Laboratory LS4FUTURE (LA/P/0087/2020); and to RL in the context of a program contract (4, 5, and 6 of article 23° of D.L. no. 57/2016 of August 29, as amended by Law no. 57/2017 of 19 July). ABB thanks ATIP-Avenir 2020 for funding. We acknowledge Ana Farinho from CEDOC's Histology Facility for assistance in tissue processing and cryosectioning. We thank Petra Pintado and Fábio Valério from CEDOC's Fish Facility for technical assistance with the support from the research consortia CONGENTO, co-financed by Lisboa Regional Operational Program (Lisboa2020), under the Portugal 2020 Partnership Agreement, LISBOA-01–0145-FEDER-022170. We also thank for technical support, Telmo Pereira from the CEDOC's Microscopy Facility, funded by PPBI-POCI-01–0145-FEDER-022122, and Claudia Andrade from the CEDOC's Flow Citometry Facility and the Instituto Gulbenkian de Ciência's Flow Cytometry.

## Additional information

### Funding

| Funder | Grant reference number | Author |
| --- | --- | --- |
| Fundação para a Ciência e a Tecnologia | PTDC/BIA-BID/29709/2017 | Ana S Brandão |
| Fundação para a Ciência e a Tecnologia | PTDC/BTM-SAL/29377/2017 | Ana S Brandão |
| Fundação para a Ciência e a Tecnologia | UIDB/04462/2020 | Ana S Brandão |
| Fundação para a Ciência e a Tecnologia | LA/P/0087/2020 | Ana S Brandão |
| Fundação para a Ciência e a Tecnologia | LISBOA-01-0145-FEDER-022170 | Ana S Brandão |
| ATIP-Avenir Program | | Anabela Bensimon-Brito |

The funders had no role in study design, data collection and interpretation, or the decision to submit the work for publication.

### Author contributions

Ana S Brandão, Conceptualization, Data curation, Formal analysis, Supervision, Validation, Investigation, Visualization, Methodology, Writing – original draft, Project administration, Writing – review and editing; Jorge Borbinha, Data curation, Formal analysis, Validation, Investigation, Visualization,

Methodology, Writing – review and editing; Telmo Pereira, Software, Formal analysis, Visualization, Methodology; Patrícia H Brito, Data curation, Software, Formal analysis; Raquel Lourenço, Investigation, Methodology, Writing – review and editing; Anabela Bensimon-Brito, Conceptualization, Writing – review and editing; Antonio Jacinto, Conceptualization, Resources, Supervision, Funding acquisition, Project administration, Writing – review and editing

#### Author ORCIDs
Ana S Brandão  http://orcid.org/0000-0001-5842-033X
Jorge Borbinha  http://orcid.org/0000-0003-0720-5642
Telmo Pereira  http://orcid.org/0000-0002-6903-9187
Raquel Lourenço  http://orcid.org/0000-0002-6176-3066
Antonio Jacinto  http://orcid.org/0000-0002-4193-6089

#### Ethics
All the people involved in animal handling and experimentation were properly trained and accredited by FELASA. All experimental procedures were approved by the Animal User and Ethical Committees at NOVA Medical School and accredited by the Portuguese National Authority for Animal Health (DGAV), according to the directives from the European Union (2010/63/UE) and Portuguese legislation (Decreto-Lei 113/2013) for animal experimentation and welfare.

#### Decision letter and Author response
Decision letter https://doi.org/10.7554/eLife.76987.sa1
Author response https://doi.org/10.7554/eLife.76987.sa2

## Additional files

#### Supplementary files
• Supplementary file 1. Spreadsheet detailing the list of significantly differentially expressed genes with p-value lower than the 0.05 threshold and log2 fold change < –1 or >1.

• Supplementary file 2. Tables detailing primer sequences for q-PCR assay (a), primary antibodies used for immunofluorescence assays(b), secondary antibodies used for immunofluorescence assays (c) and sample size number and statistical test preformed for each quantitative experimental design (d).

• Transparent reporting form

#### Data availability
Microarray data have been deposited in GEO under accession code GSE194385.

The following dataset was generated:

| Author(s) | Year | Dataset title | Dataset URL | Database and Identifier |
|---|---|---|---|---|
| Brandão AS, Brito PH | 2022 | Zebrafish caudal fin amputation induces a metabolic switch necessary for cell identity transitions and cell cycle re-entry to support blastema formation and bone regeneration | https://www.ncbi.nlm.nih.gov/geo/query/acc.cgi?acc=GSE194385 | NCBI Gene Expression Omnibus, GSE194385 |

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
