## [Editor Report]

The authors provide convincing evidence to show that injury induces activation of glycolysis during zebrafish adult tail fin regeneration. This early activation is crucial for osteoblast dedifferentiation and proliferation, which are required for blastema formation and tail fin regeneration. This important study will be of interest to a broad audience in the fields of regeneration and metabolic regulation of developmental processes.

---

## [Decision Letter]

**Decision letter after peer review:**

Thank you for submitting your article "Zebrafish caudal fin amputation induces a metabolic switch necessary for cell identity transitions and cell cycle re-entry to support blastema formation and bone regeneration" for consideration by *eLife*. Your article has been reviewed by 2 peer reviewers, and the evaluation has been overseen by Phillip Newmark as Reviewing Editor and Didier Stainier as the Senior Editor. The following individual involved in review of your submission has agreed to reveal their identity: Yi Feng (Reviewer #2).

Essential revisions:

1) One of the major claims of the manuscript is that there is a "metabolic switch" from OXPHOS to aerobic glycolysis. The authors clearly show that enhanced glycolysis is indispensable for osteoblast dedifferentiation and cell cycle reentry during regeneration; however, they do not provide direct evidence that OXPHOS is diminished. Although treatment with two OXPHOS inhibitors (MB6, UK5099) does not inhibit regeneration, the authors provide no rationale or citations for the concentrations of the drugs used here, nor validation that these concentrations are effective in the animal. Thus, for the authors to maintain their assertion of a "metabolic switch," they should provide further evidence to support the downregulation of OXPHOS (e.g., is PDK upregulated? Is mitochondria pyruvate carrier downregulated? Are TCA cycle intermediates downregulated?). In the absence of such additional support for OXPHOS downregulation, the authors should modify their text (title, abstract, and main manuscript) and tone down their conclusions accordingly.

2) In the course of our editorial consultation, one reviewer noted that if UK5099 treatment blocks pyruvate entry into mitochondria, it should lead to enhanced lactate production. Based on the authors' model, enhanced lactate might be predicted to enhance regeneration. Figure 3S shows a slightly increased blastema area in UK5099-treated animals, but the sample size is quite small (n=5). The authors are encouraged to repeat this experiment and increase their sample sizes.

3) In their comments (included below), reviewer #1 noted that the terminology used in this manuscript to define proximal and distal blastema, as well as the patterning zone is not consistent with standard usage in the field. The figures and text should be modified so that use of these terms is consistent with standard usage. The authors should also comment on why they appear to be detecting proliferation in the distal blastema (Figures 6H, S6A-C), in contrast to previous reports.

4) As requested by Reviewer #2 in their public review, please provide more details of the microarray analysis reported in Figure S1, along with documentation of the differentially expressed genes, including annotations, fold changes, and p values.

*Reviewer #1 (Recommendations for the authors):*

1. Positive impact of glycolysis activation or oxidative phosphorylation (OXPHOS) inhibition

The authors nicely demonstrate the negative impact of glycolysis inhibition on fin regeneration and its cellular mechanisms, such as dedifferentiation, blastema formation and proliferation. However, the authors did not address another important aspect whether activation of glycolysis or inhibition of OXPHOS can enhance regeneration. One can argue that glycolysis is an essential metabolism and thus inhibition of glycolysis may cause defects of any biological events, including regeneration. Thus, another important approach is to determine whether metabolic reprogramming can enhance the regenerative program. I understand that the authors tried to address this by targeting OXPHOS but OXPHOS enzymes do not show differential transcription and UK5099 and MB6 do not impact fin regeneration. To address this further, I recommend the following:

1) LC-MS analysis

It is unclear whether the authors target to profile only four metabolites with LC-MS analysis (Figure 2E). I expect that the authors may be able to get more data from LC-MS, such as a profile of TCA cycle intermediates and key metabolites (succinate, citrate, malate, and so on). Even, there are more glucose metabolites, such as glucose-6-phosphate, fructose 6-phosphate, and so on. As the authors focus on glycolysis and OXPHOS, the profiles of more glycolysis and TCA cycle metabolite changes should be provided.

2) Functional impact of glycolysis activation and/or OXPHOS inhibition

The authors used two drugs, MB6 and UK5099. However, it is unclear whether treatment of these drugs can efficiently and specifically inhibit OXPHOS. By brief literature search, it is hard to find out papers using these two drugs with animals (although there are multiple papers treating these drugs with cells). Thus, I recommend alternative approaches.

a) Genetic method

Fukuda et al., (2020, EMBO reports, DOI 10.15252/embr.201949752) employed genetic manipulation to enhance glycolysis in adult zebrafish. They used ppargc1a mutants and pdk3b overexpression line, providing evidence that metabolic switch can result in positive effects of heart regeneration. Magadum et al., (2020, Circulation, DOI: 10.1161/CIRCULATIONAHA.119.043067) also used mouse genetics to transiently express Pkm2 and demonstrated positive effects on heart regeneration. Note that pkm2 is a well-characterized glycolytic enzyme in cancer that promotes glucose metabolism toward lactate. Based on Fukuda's work, pkma2 is likely a homolog of pkm2 in zebrafish. The authors consider using one of these animal models to determine whether metabolic reprogramming can enhance fin regeneration.

b) Pharmaceutical approach

An alternative approach is to change metabolism by treating drugs. Bae et al., (2021, Circulation, DOI: 10.1161/CIRCULATIONAHA.120.049952) changed mouse metabolism by treating several drugs. They inhibit the succinate dehydrogenase enzyme complex by treating malonate and Atpenin A5 in mice. The authors consider treating these drugs and assess whether the metabolic switch can enhance fin regeneration.

2. Blastema subdivision

The blastema is defined as a proliferative cell mass, indicating that there are dedifferentiated cells, and once cells exit blastema they differentiate. In a 2002 paper (Nechiporuk and Keating, Development), the authors identified there are two distinct populations in the blastema. Distally located blastema cells are non-proliferative but express blastema marker msxb, naming it as distal blastema (DB). Thereafter, this DB is similarly used by multiple groups in the fin regeneration field as shown by several papers (Kang et al., Dev. Cell, 2013; Wehner et al., Cell Reports, 2014). This DB is a very small domain at the tip of mesenchymal cells. Based on previous works (Wehner et al., Cell Reports, 2014 and Stewart et al., Cell Reports, 2014), this DB is not marked by runx2 positive cell lining. By contrast, proximal blastema (PB) contains bilateral zones of proliferative pre-OB runx2 positive cell layers. As PB is a major blastema region and comprises proliferative cells, the PB ends by emerging differentiated cells, such as osterix positive cells. As patterning zone (PZ) indicates differentiation, the PZ area starts with osx expressing cells. This compartmentalization of blastema is standard in fin regeneration. However, the authors define DB, PB, and PZ incorrectly. In Figure 6B, DB actually indicates PB and both PB and PZ indicate PZ. DB is a very small domain distal to PB, which is not annotated in Figure 6B. Thus, Figure 6 and Sup Figure 5 use incorrect indications and result 5 "Glycolysis suppression leads to ~" is also incorrectly described.

Line 321 "show cycling Runx2+ pre-OBs maintain the progenitor pool in the DB"

– This is not DB, but PB.

Line 322" as they proliferate and populate the PB, they differentiate into fast-proliferating Runx2+ Osx+ immature OBs"

– This describes PZ, not PB (or the border of PB and PZ as it is not possible to clearly subdivide these two regions).

I highly recommend revising the manuscript (result section 5 and discussion) and figures based on the definition of PB and DB widely used in the field.

3. Long-term effects of glycolysis inhibition

The authors assess fin regeneration before 48hpa. How about after 48 hours? Are there any bone phenotypes at 4 or 7 dpa or is bone completely lost?

4. Effect of glycolysis inhibition on uninjured fins (homeostasis)

Is there any outcome on OB behavior or fin integrity from blocking glycolysis (with 2DG) in uninjured fins?

---

## [Author Response]

Essential Revisions (for the authors):1) One of the major claims of the manuscript is that there is a "metabolic switch" from OXPHOS to aerobic glycolysis. The authors clearly show that enhanced glycolysis is indispensable for osteoblast dedifferentiation and cell cycle reentry during regeneration; however, they do not provide direct evidence that OXPHOS is diminished. Although treatment with two OXPHOS inhibitors (MB6, UK5099) does not inhibit regeneration, the authors provide no rationale or citations for the concentrations of the drugs used here, nor validation that these concentrations are effective in the animal. Thus, for the authors to maintain their assertion of a "metabolic switch," they should provide further evidence to support the downregulation of OXPHOS (e.g., is PDK upregulated? Is mitochondria pyruvate carrier downregulated? Are TCA cycle intermediates downregulated?). In the absence of such additional support for OXPHOS downregulation, the authors should modify their text (title, abstract, and main manuscript) and tone down their conclusions accordingly.

We appreciate the reviewers concerns and feedback on this matter and agree that in order to use the term “metabolic switch” we need to provide further evidence. We have followed the reviewer’s suggestions and our results were the following:

– By evaluating the expression profile of *pdk*, the inhibitor of pdh, namely *pdk3a* and *pdk3b*, we observe that there is an actual downregulation of the expression of both genes, which goes against a downregulation of OXPHOS. However, we highlight that in our data (Figure 2C) we also observe a clear downregulation of *pdha1b*, which converts pyruvate into acetyl-CoA, at 6 hpa when changes in energy metabolism start to occur. The expression profile of the mitochondria pyruvate carriers analysed, *mpc1* and *mpc2*, was unchanged at 6 and 24 hpa. Overall, the analysis by qPCR of the expression of both *pdk* and *mpc* components does not provide clear evidence for a “metabolic switch”. Given this, we decided to not include these data into the manuscript, but we provide the reviewers the graphs containing the data mentioned above.

**Author response image 1. sa2fig1:** Relative gene expression of *mpc1, mpc2, pdk3b*, in the whole caudal fin stump, a (A) 6hpa and at (B) 24 hpa in comparison to uninjured conditions (0 hpa). Statisical analysis with *t* test (n=5 (A) and 4 (B) biological replicates).

– To assess TCA cycle intermediates, which could give further proof of a “metabolic switch” during fin regeneration, we did an untargeted MS protocol at 6 and 24 hpa in relation to 0 hpa. Unfortunately, for our 6 and 24 hpa samples, the amount of most of the metabolites was very low, bellow the MS instrument sensibility, which limited the data that we could analyse and that we were confident enough to include in the manuscript. Since this experiment is very time consuming and requires many animals, we could not repeat the assay. We were only able to use the measurements for Citrate, the first metabolic intermediate from TCA cycle generated from acetyl-CoA, and α-KG for the 6 hpa time-point. Citrate is significantly downregulated, and α-KG remained unchanged, which suggests a lower contribution of acetyl-CoA to fuel TCA cycle and be converted into Citrate, thus potentially reducing of OXPHOS. Given that few TCA intermediates were retrieved from our MS assay, we were not able to further improve our arguments on this point. We decided to not include these results into the manuscript but provide the data to the reviewers in this document.

**Author response image 2. sa2fig2:** Relative measurement of Citrate and a-KG metabolite levels, in the whole caudel fin stump, at 6 hpa in comparison to uninjured conditions (0 hpa) Statistical analysis with Turkey HSD test (n=4 biological replicates). Mean and SD displayed on the graphs. *n.s., not significative, *, p-value <0,01*.

Regarding the reviewer’s comment on, the drugs used in this manuscript to inhibit OXPHOS, UK5099 and MB6, our rationale to the amount of drug administered via IP was the following:

“For UK5099, we used the higher concentration that allows a proper dissolution of the compound in the solvent used (PBS:DMSO, 1:1 mixture). This concentration did not lead to mortality nor signs of toxicity in the animals used in our study. This pharmacological compound was previously validated and shown to work in mice for inhibition of mitochondrial OXPHOS (Corbet et al., 2018; Buyse et al., 2021).

For MB6, animals were incubated with an already validated concentration, previously described in two studies using zebrafish larvae (Dabir et al., 2013; Sinclair et al., 2021). In our setting, animals incubated with the same concentration of MB6 did not show signs of toxicity nor increase in mortality.”

We hope this explanation helps clarifying this point.

In addition to the suggested experiments, we also looked at mitochondria dynamics in this context. Usually, changes in cellular metabolism are accompanied and are highly influenced by alterations in the mitochondria morphology, shape and size, due to fusion and fission events or even due to mitochondria biogenesis (Wai and Langer, 2016). In fact, there are many studies linking mitochondria morphology and function (Zemirli et al., 2018). In general terms, while differentiated cells possess a fused and elongated mitochondria network that sustains OXPHOS, the mitochondria of stem cells or proliferating cells, which rely on glycolysis, are smaller and spherical due to increase in fission events (Wai and Langer, 2016). Here, using a reporter line that labels mitochondria, MLS-GFP, we made two important observations that suggest that changes in metabolism occur in parallel or are accompanied by an increase in fission events at 6 hpa:

– Genes responsible for mitochondria fission, like *drp1* and *fis1,* were upregulated at 6 hpa;

– By analysing the number and size of mitochondria at 0, 6 and 24 hpa, we observed that there is an increase in mitochondria number at 6 and 24 hpa, when compared to uninjured caudal fins (0 hpa). Strikingly, at 6 hpa there is an increase in the percentage of smaller mitochondria (<0.1 µm^3^), which is not observed at 24 hpa.

These data suggest that early stages of caudal fin regeneration, at 6 hpa time-point, are characterized by an enhancement of glycolysis that is associated with a potential increase of mitochondria fission. The increase in mitochondrial number and percentage of smaller mitochondria is indicative of mitochondria fission that, in other contexts, is known to be correlated with decrease of mitochondrial OXPHOS activity. Furthermore, similar observations were shown to occur during zebrafish heart regeneration (Honkoop et al., 2019). The authors demonstrated that border zone cardiomyocytes near the injury site, were characterized by having smaller and immature mitochondria, suggesting a reduced OXPHOS activity. Likewise, our data seems to support a tendency to decreased mitochondrial OXPHOS, at least at 6 hpa, however, a more detailed analysis on mitochondrial dynamics using other tools and evaluating other markers related to mitochondria fusion should be done to further support the data. Although mitochondria dynamics was not the primary focus of this work, these experiments presented very interesting results that could further support the manuscript’s main idea. Thus, we decided to add this data to the manuscript on new Figure 2—figure supplement 2 (Lines 212-232 and 425-426 in the main text, and Lines 587; 767-769; 814-818 in Material and Methods) and expect that these results contribute to clarifying the main message of this work.

Taking into account all the comments above and after carefully analysing our new sets of data we consider that “metabolic switch” may be a strong term to use in this case, so we opted to tone down our conclusions, as the reviewers also suggested. Therefore, we removed “metabolic switch” and rephrased the title and all the sentences throughout the main text and figures and, instead, we propose that our data is indicative of a metabolic adaptation that increases glycolysis, occurring at early stages of regeneration and supporting regeneration. We hope the reviewers agree that these modifications make the manuscript more precise and, importantly, are in accordance with the experimental data.

2) In the course of our editorial consultation, one reviewer noted that if UK5099 treatment blocks pyruvate entry into mitochondria, it should lead to enhanced lactate production. Based on the authors' model, enhanced lactate might be predicted to enhance regeneration. Figure 3S shows a slightly increased blastema area in UK5099-treated animals, but the sample size is quite small (n=5). The authors are encouraged to repeat this experiment and increase their sample sizes.

We thank the reviewer for making this observation. As mentioned, the sample size used for this experiment was quite low, which may have masked an effect of UK5099 to increase regeneration. By following the reviewer’s suggestion and increasing the sample size, we observed that, in fact, administration of UK5099 lead to a slight but significant increase of the regenerated area (Figure 3P-S). This indicates that blocking pyruvate entry into the mitochondria could potentiate pyruvate conversion into lactate, which we have shown to be important for proper regeneration, enhancing this branch of glycolysis. These data are also in accordance with a recent study in zebrafish showing that stimulation of glycolysis can promote proliferation and regeneration (Fukuda et al., 2019). This new set of results was added to the manuscript on Figure 3S (Lines 270-275 in the main text). We expect this data to make the manuscript more interesting and exciting and thank the reviewer for leading us in this direction.

3) In their comments (included below), reviewer #1 noted that the terminology used in this manuscript to define proximal and distal blastema, as well as the patterning zone is not consistent with standard usage in the field. The figures and text should be modified so that use of these terms is consistent with standard usage. The authors should also comment on why they appear to be detecting proliferation in the distal blastema (Figures 6H, S6A-C), in contrast to previous reports.

The reviewer #1 is correct, the definitions of the PZ, PB and DB are inconsistent with the terminology used in the field. We truly apologize for this oversight in our study and thank reviewer #1 for his remark. To be in accordance with the standard nomenclature used for the blastema subdivision, we have rectified and modified these terms to the proper definition in the main text (Lines 351-356 and 366 in the main text) and blastema representation in Figure 6B. In addition, we decided to withdraw blastema compartmentalization from images of caudal fin cryosections as the boundaries between the different compartments could be difficult to define, namely between the PB and PZ.

The reviewer also spotted dividing cells within the DB compartment (Figure 6H and S6A-C (now Figure 6—figure supplement 2A-C)) and expresses concerns, as DB cells are described to be non-proliferative (Nechiporuk and Keating, 2002). There are two main reasons for detecting cell proliferation within the DB:

– The initial reports addressing cell proliferation used mainly BrdU (Nechiporuk and Keating, 2002) and we used EdU, which could lead to differences in cell labelling and variances in the number of cells being detected. EdU is a simpler, quicker and sensitive assay, in contrast to BrdU, which requires harsh treatments that can compromising tissue integrity. Some studies observed discrepancies using the two methodologies and suggested that for the EdU and BrdU treatments to be considered equivalent, the exposure time to BrdU should be increased (da Silva et al., 2017). Thus, there may also exist differences between these two methods in the context of caudal fin regeneration that were not addressed before and should be potentially reviewed in the future;

– Although DB cells are characterized to be non-proliferative (Nechiporuk and Keating, 2002), depending on the BrdU incorporation interval there are cell within the DB demonstrating proliferative capacity. As quoted from the Nechiporuk and Keating, 2002 study: “During various lengths of BrdU treatment, we noticed that the number of proximally labeled DMB cells progressively increased with longer treatments (Figure 5C,E,F), suggesting that proximal DMB cells cycle at a slow rate".

The authors use several BrdU treatments to assess proliferation in the DB, 40min, 140min and 6 h, showing a correlation between the duration of treatment and the increase in number of proliferating cells, suggesting that DM is composed of slow cycling cells. Since we perform a 3 hour EdU incorporation assay, we could be detecting these slow cycling DB cells.

Therefore, our interpretation and explanation for detecting proliferative cells in the DB regards differences in methodology used to label proliferative cells. Due to differences in the compounds used to label proliferative cells and the incorporation time-window, these methodologies cannot be considered equivalent. We hope to have addressed the reviewer concerns on this point.

4) As requested by Reviewer #2 in their public review, please provide more details of the microarray analysis reported in Figure S1, along with documentation of the differentially expressed genes, including annotations, fold changes, and p values.

As requested by reviewer #2, we now supply more details regarding the differential expressed genes obtained in the osteoblastArrayXS. In the Methods section, we added more information on the methods used to infer significantly differential expression (Lines 711-715 in Material and Methods). As mentioned in the main text, we submitted to NCBI Gene Expression Omnibus archive the transcriptome datasets analysed on this study (accession number GSE194385). Nevertheless, we now follow the reviewer suggestion and provide expression values for all samples and a list of the differentially expressed genes including annotations, fold changes and p values. This additional information is provided in Supplementary File 1. Please, also note that the data provided in the main Figure 2B regards a set of differentially expressed genes retrieved from the osteoblastArrayXS, which illustrates the differences in some glycolytic enzymes and OXPHOS components at 6 hpa in relation to uninjured (0 hpa). To make sure that this is clear on the text, we have rephrased it in the revisited manuscript (Lines 168-170 in the main text). By providing this additional data we give a more complete picture on metabolism related genes and pathways important for osteoblast dedifferentiation.

Reviewer #1 (Recommendations for the authors):1. Positive impact of glycolysis activation or oxidative phosphorylation (OXPHOS) inhibitionThe authors nicely demonstrate the negative impact of glycolysis inhibition on fin regeneration and its cellular mechanisms, such as dedifferentiation, blastema formation and proliferation. However, the authors did not address another important aspect whether activation of glycolysis or inhibition of OXPHOS can enhance regeneration. One can argue that glycolysis is an essential metabolism and thus inhibition of glycolysis may cause defects of any biological events, including regeneration. Thus, another important approach is to determine whether metabolic reprogramming can enhance the regenerative program. I understand that the authors tried to address this by targeting OXPHOS but OXPHOS enzymes do not show differential transcription and UK5099 and MB6 do not impact fin regeneration. To address this further, I recommend the following:1) LC-MS analysisIt is unclear whether the authors target to profile only four metabolites with LC-MS analysis (Figure 2E). I expect that the authors may be able to get more data from LC-MS, such as a profile of TCA cycle intermediates and key metabolites (succinate, citrate, malate, and so on). Even, there are more glucose metabolites, such as glucose-6-phosphate, fructose 6-phosphate, and so on. As the authors focus on glycolysis and OXPHOS, the profiles of more glycolysis and TCA cycle metabolite changes should be provided.

We thank the reviewer for the pertinent observation. To address the reviewer suggestion, we have tried to perform an untargeted MS analysis on caudal fin from 0, 6 and 24 hpa to assess whether TCA cycle intermediates were downregulated at 6 hpa and 24 hpa, in comparison to 0 hpa. As mentioned above, many TCA cycle metabolites at 6 and 24 hpa were not suitable for analysis since they did not pass the quality control, potentially due to the low amount of the metabolites obtained in these conditions. Unfortunately, we were only able to properly measure Citrate and α-KG at 6 hpa in relation to uninjured fins (0 hpa). Given that this is a very time-consuming experiment that requires many animals, we could not repeat the MS assay. Thus, the analysis of TCA cycle intermediates remains incomplete. As an alternative, we also tried, by other means, to demonstrate that there is an increase in glycolysis and a diminished OXPHOS activity by addressing mitochondria dynamics in this context. Usually, changes in cellular metabolism are accompanied and are highly influenced by alterations in the mitochondria morphology, shape and size due to fusion and fission events or even due to mitochondria biogenesis (Wai and Langer, 2016). These changes in morphology and intimately correlated to mitochondria function (Zemirli et al., 2018). In general terms, while differentiated cells possess a fused and elongated mitochondria network that sustains OXPHOS, the mitochondria of stem cells or proliferating cells which rely on glycolysis, are smaller and spherical due to increase in fission events (Wai and Langer, 2016). Using a reporter line that labels mitochondria, MLS-GFP, we made two important observations that suggest that changes in metabolism occur in parallel or are accompanied by an increase in fission events at 6 hpa:

– Genes responsible for mitochondria fission, like *drp1* and *fis1,* were upregulated at 6 hpa;

– By analysing the number and size of mitochondria at 0, 6 and 24 hpa, we observed that there is an increase in mitochondria number at 6 and 24 hpa, when compared to uninjured caudal fins (0 hpa). Strikingly, at 6 hpa there is an increase in the percentage of smaller mitochondria (<0.1 µm^3^) which is not observed at 24 hpa.

This suggests that early stages of caudal fin regeneration, at 6 hpa time-point, are characterized by an increase of mitochondria fission that is related to the increase in glycolysis and can potentially lead to a decrease in OXPHOS activity. Furthermore, similar observations were shown to occur during zebrafish heart regeneration (Honkoop et al., 2019). The authors demonstrated that border zone cardiomyocytes near the injury site, were characterized by having smaller and immature mitochondria, suggesting a reduced OXPHOS activity. Likewise, our data could support that, at least at 6 hpa, there could be a tendency to decrease mitochondrial OXPHOS, however a more detailed analysis on mitochondrial dynamics using other tools and evaluation of other markers related to mitochondria fusion should be done to further support the data. Although mitochondria dynamics was not the primary focus of this work, these experiments showed very interesting results that could further support the manuscript’s main idea. This data was added to the manuscript on new Figure 2—figure supplement 2 (Lines 212-232 and 425-426 in the main text, and Lines 587; 767-769; 814-818 in Material and Methods).

After carefully analysing the data, we think that there is still some doubt related to the role of OXPHOS and whether it is really being suppressed after caudal fin amputation. Therefore, we decided to be more conservative regarding our conclusions. Instead of proposing that a complete “metabolic switch” occurs during fin regeneration, we rephrased and changed the main manuscript and figures so that the main idea is that there is a metabolic adaptation in the form of an increase in glycolysis and in the glycolytic influx that supports regeneration, not excluding a potential effect or role of OXPHOS. We acknowledge that by doing those modifications the manuscript becomes clearer, more accurate and precise and, importantly, in accordance with the experimental data.

2) Functional impact of glycolysis activation and/or OXPHOS inhibitionThe authors used two drugs, MB6 and UK5099. However, it is unclear whether treatment of these drugs can efficiently and specifically inhibit OXPHOS. By brief literature search, it is hard to find out papers using these two drugs with animals (although there are multiple papers treating these drugs with cells). Thus, I recommend alternative approaches.

We understand the reviewer relevant comments and concerns and acknowledge that further information should be provided regarding the drugs used in this study. We also thank the reviewer for providing the following alternative suggestions. Regarding the reviewer’s apprehension regarding the drugs used in this manuscript to inhibit OXPHOS, UK5099 and MB6, our rationale to the amount of drug administered via IP was the following:

– For UK5099, we used the higher concentration that allows a proper dissolution of the compound in the solvent used (PBS:DMSO, 1:1 mixture). This concentration did not lead to mortality nor signs of toxicity in the animals used in our study. This pharmacological compound was previously validated and shown to work in mice for inhibition of mitochondrial OXPHOS (Corbet et al., 2018; Buyse et al., 2021).

– For MB6, animals were incubated with an already validated concentration, previously described in two studies using zebrafish larvae (Dabir et al., 2013; Sinclair et al., 2021). In our setting, animals incubated with the same concentration of MB6 did not show signs of toxicity nor increase in mortality.

We hope this explanation helps clarifying this point.

a) Genetic methodFukuda et al., (2020, EMBO reports, DOI 10.15252/embr.201949752) employed genetic manipulation to enhance glycolysis in adult zebrafish. They used ppargc1a mutants and pdk3b overexpression line, providing evidence that metabolic switch can result in positive effects of heart regeneration. Magadum et al., (2020, Circulation, DOI: 10.1161/CIRCULATIONAHA.119.043067) also used mouse genetics to transiently express Pkm2 and demonstrated positive effects on heart regeneration. Note that pkm2 is a well-characterized glycolytic enzyme in cancer that promotes glucose metabolism toward lactate. Based on Fukuda's work, pkma2 is likely a homolog of pkm2 in zebrafish. The authors consider using one of these animal models to determine whether metabolic reprogramming can enhance fin regeneration.

We fully understand that genetics would be the best approach to further explore our data. However, when Fukuda et al., (2020, EMBO reports) was published, we already had most of the data regarding 2DG, UK5099 and SO. We have considered to use *ppargc1a* and *pkma2^-/-^; pkmb^-/-^* mutants, which were the more suitable to us and for the adult zebrafish studies. The use of these lines could confirm our results and possibly demonstrate show that fin regeneration is improved by stimulating glycolysis. Nevertheless, this would increase greatly the time required to perform the experiments This would require permission to perform a new set of experiments with animals, and mutant and sibling fish to grow into adulthood (approximately 3/4 months) to be genotyped and then subjected to experimentation. Alternatively, as mentioned above in the “Essential Revisions” section, we have increased the sample size for the UK5099 treatment, which, since it blocks pyruvate transportation to the mitochondria, may enhance pyruvate conversion into lactate and potentiate aerobic glycolysis. We observed that administration of UK5099 lead to a slight but significant increase of the regenerated area (Figure 3P-S). This indicates that blocking pyruvate entry into the mitochondria could potentiate pyruvate conversion into lactate, which we have shown to be also important for proper regeneration, enhancing this branch of glycolysis. We hope this new set of data is enough to persuade the reviewer.

b) Pharmaceutical approachAn alternative approach is to change metabolism by treating drugs. Bae et al., (2021, Circulation, DOI: 10.1161/CIRCULATIONAHA.120.049952) changed mouse metabolism by treating several drugs. They inhibit the succinate dehydrogenase enzyme complex by treating malonate and Atpenin A5 in mice. The authors consider treating these drugs and assess whether the metabolic switch can enhance fin regeneration.

We thank the reviewer for his/her suggestion. We would like to avoid using another drug since the process of optimization could be very time-consuming and titration is usually required. Also, the drugs used in this manuscript, UK5099 and MB6, have been shown to block the oxidative branch of glycolysis, both in mice and in zebrafish (i.e., MB6). In addition, we also think that using UK5099 is a more effective approach to test whether the metabolic adaptation can enhance fin regeneration, since it inhibits the pyruvate entry into mitochondria, thereby pyruvate is available to fuel, support and increase aerobic glycolysis. As suggested by the reviewer, inhibition of the mitochondrial succinate dehydrogenase enzymatic complex, part of the electron transport chain, would allow to access whether OXPHOS is important to support regeneration, similar to the MB6 context. Importantly, as suggested by the reviewer, we also test the ability of UK5099 to enhance of caudal fin regeneration and in fact we observed an increase in the regenerative area of the fin at 48 hpa. Given this new set of results, we became more confident that UK5099, potentially by enhancing glycolysis, to have a beneficial, although modest, effect on fin regeneration. We hope this explanation and the new set of result obtained with UK5099 is sufficient to clarify this point and these arguments can convince the reviewer that UK5099 is the more suitable compound to address this point.

2. Blastema subdivisionThe blastema is defined as a proliferative cell mass, indicating that there are dedifferentiated cells, and once cells exit blastema they differentiate. In a 2002 paper (Nechiporuk and Keating, Development), the authors identified there are two distinct populations in the blastema. Distally located blastema cells are non-proliferative but express blastema marker msxb, naming it as distal blastema (DB). Thereafter, this DB is similarly used by multiple groups in the fin regeneration field as shown by several papers (Kang et al., Dev. Cell, 2013; Wehner et al., Cell Reports, 2014). This DB is a very small domain at the tip of mesenchymal cells. Based on previous works (Wehner et al., Cell Reports, 2014 and Stewart et al., Cell Reports, 2014), this DB is not marked by runx2 positive cell lining. By contrast, proximal blastema (PB) contains bilateral zones of proliferative pre-OB runx2 positive cell layers. As PB is a major blastema region and comprises proliferative cells, the PB ends by emerging differentiated cells, such as osterix positive cells. As patterning zone (PZ) indicates differentiation, the PZ area starts with osx expressing cells. This compartmentalization of blastema is standard in fin regeneration. However, the authors define DB, PB, and PZ incorrectly. In Figure 6B, DB actually indicates PB and both PB and PZ indicate PZ. DB is a very small domain distal to PB, which is not annotated in Figure 6B. Thus, Figure 6 and Sup Figure 5 use incorrect indications and result 5 "Glycolysis suppression leads to ~" is also incorrectly described.Line 321 "show cycling Runx2+ pre-OBs maintain the progenitor pool in the DB"– This is not DB, but PB.Line 322" as they proliferate and populate the PB, they differentiate into fast-proliferating Runx2+ Osx+ immature OBs"– This describes PZ, not PB (or the border of PB and PZ as it is not possible to clearly subdivide these two regions).I highly recommend revising the manuscript (result section 5 and discussion) and figures based on the definition of PB and DB widely used in the field.

The reviewer is absolutely correct, the definitions of the PZ, PB and DB are inconsistent with the terminology used in the field and therefore are not correct. We truly apologize for this flaw in our study and thank reviewer #1 for noticing this error. In accordance with the nomenclature used to define the blastema regions, we have rectified and modified these terms to the proper definition in the main text (Lines 351-356 and 366 in the main text) and representation in Figure 6B. We decided to withdraw blastema compartmentalization from images of caudal fin cryosections as the boundaries between the different compartments could be difficult to define, namely between the PB and PZ. We hope that the changes made to this revisited version of the manuscript are now correct regarding blastema nomenclature.

3. Long-term effects of glycolysis inhibitionThe authors assess fin regeneration before 48hpa. How about after 48 hours? Are there any bone phenotypes at 4 or 7 dpa or is bone completely lost?

We performed these experiments as the reviewer suggested. However, we had to do slight changes to the protocol. Instead of performing 2 IP injections per day (every 12 h) with PBS or 2DG (as represented on Figure 3B), we opted to give only a single injection of PBS or 2DG per day. The reason for this is that 7 days of treatment with a dose of 2DG every 12h, lead to an increase in toxicity and mortality in 2DG-treated fish. Therefore, by using this extended protocol until 7 dpa with a single PBS or 2DG injection per day, we were able to avoid mortality and visible toxic effects and, importantly, we were able to observe that bone regeneration is completely abolished in 2DG-trested animals in comparison to control animals. This indicates that 2DG is fundamental for regeneration and bone formation. This new data was added to the main manuscript (Lines 392-396 in the main text, and Lines 632-637 in Material and Methods) and to Figure 6—figure supplement 4A,C-F. We hope the new data on this matter is sufficient to clarify the role of glycolysis during bony-ray regeneration.

4. Effect of glycolysis inhibition on uninjured fins (homeostasis)Is there any outcome on OB behavior or fin integrity from blocking glycolysis (with 2DG) in uninjured fins?

We thank the reviewer for her/his suggestions and have performed the experiments to address the effect of glycolysis inhibition during homeostasis. We opted to use the same protocol used above to address point 3 of reviewer#1. We did single IP injections of PBS or 2DG during 7 consecutive days on fish with uninjured caudal fins. After the 7 days, fins were collected and stained fins for Calcein. We measured general and simple parameters that can be used as a readout of fin integrity, like fin area to width ratio and bony-ray length to width ratio. We did not detect differences in those parameters or any obvious malformations, suggesting that inhibition of glycolysis does not seem to impact fin homeostasis, but as a crucial role in controlling regeneration. These data were added to the main manuscript (Lines 396-401 in the main text, and Lines 632-637 in Material and Methods) and to Figure 6—figure supplement 4B,G-N. We hope this data helps clarifying this point.